# SWE-Bench Atlas: A Framework for the Scalable Generation of Software Engineering Benchmark from Open-Source Repositories

## Abstract

Benchmarks like SWE-bench have shaped the evaluation of Large Language Models (LLMs) on complex software engineering tasks. However, these efforts remain limited by manual curation, static datasets, and a focus on Python-based bug fixes. We introduce SWE-Bench Atlas, a fully automated framework for generating high-fidelity, large-scale, multilingual, and diverse real-world repository-level coding tasks from open-source GitHub projects.Unlike synthetic frameworks or manually curated sets, SWE-Bench Atlas introduces an end-to-end pipeline that continuously harvests live pull requests to capture a broad spectrum of real-world software engineering demands, including both bug fixes and feature requests. The framework operates via a five-stage automated pipeline: (1) a Sourcing Module that identifies high-quality pull requests across diverse languages; (2) a Neuro-Symbolic Dockerization System that utilizes tool-augmented and template-guided synthesis to enforce strict reproducibility; (3) a State-Differential Test Oracle Extraction that integrates Adaptive Log Parsing to verify both regressions and feature requests across heterogeneous build system; (4) an Automated Quality Assurance to ensure environmental determinism; and (5) a Hint-Guided Trajectory Synthesis module that converts model-breaking instances into high-value training data. Our initial benchmark consists of 11,133 instances from 3,971 repositories across 11 languages. On a subset of 1,782 instances of this benchmark, today's strongest models perform as follows: `claude-sonnet-4.5` (36.20% pass@10), `gpt-5-2025-08-07` (34.57%), `gemini/gemini-2.5-pro` (16.89%), and `gpt-4o` (18.24%). Supplementing our findings, a public release of 500-task dataset, together with evaluation scripts is included in the supplementary material. We further demonstrate the utility of our dataset by showing that fine-tuning on SWE-Bench Atlas instances yields measurable improvements on the SWE-bench Multilingual benchmark. By automatically producing dynamic, polyglot, and verifiable tasks, SWE-Bench Atlas enables scalable evaluation and advancement of AI coding and reasoning abilities of next-generation AI systems.

## 1 Introduction: The Need for Scalable, Dynamic Software Engineering Benchmarks

The evaluation of Large Language Model (LLM) coding agents has shifted from isolated function synthesis (e.g. HumanEval (Chen et al., 2021)) to repository-level software engineering. **SWE-bench** established a novel methodology for this setting by evaluating agents on real-world GitHub issues, establishing a gold standard for realism (Jimenez et al., 2024). However, its reliance on manual curation restricts its scope to just 12 Python repositories, a sample size that fails to capture the structural and linguistic diversity of the open-source ecosystem.

Attempts to address these scalability limitations have followed two distinct but insufficient paths:

1. **Manual Multilingual Expansion:** Benchmarks such as **Multi-SWE-bench** (Zan et al., 2025) and **SWE-bench Multilingual** (Yang et al., 2025) extended the evaluation framework to languages like Java and Rust. However, these efforts rely on manual curation, limiting their scope to a few dozen repositories and preventing the statistical power required for robust evaluation.

2. **Python-Centric Automation:** SWEE-bench introduced automation to the pipeline, scaling coverage to hundreds of repositories (Vergopoulos et al., 2025). However, it remains restricted to Python and suffers from two critical technical limitations. First, its two-state test oracle (Before-patch → After-patch) is not designed to extract feature requests that introduce new APIs or functionality, as these cause the Before

state to fail to build due to missing symbols—cases that existing pipelines must filter out as errors. This methodological constraint restricts automated benchmarks to scenarios where tests can execute in both states, severely limiting feature request coverage. Second, it relies on static regular expressions for log parsing, preventing it from scaling to the "long tail" of repositories with heterogeneous test runners and non-standard outputs.

Furthermore, recent works like **SWE-Smith** (Yang et al., 2025) and **SWE-Flow** (Zhang et al., 2025a) have attempted to scale data generation via synthetic means, such as synthesizing tasks from Test-Driven Development (TDD) patterns. While valuable for training, these synthetic approaches do not serve the primary goal of evaluation on "in-the-wild" distributions. They lack the noisy, complex, and historical nature of human-written code. Additionally, the static nature of all aforementioned benchmarks introduces a critical **data contamination risk**: most instances were created before the training cutoff of modern models, rendering them prone to memorization.

To bridge these gaps, we present **SWE-Bench Atlas**, the first fully automated, multilingual framework designed to continuously expand the horizon of software engineering benchmarks. Unlike previous approaches, our methodology provides a systematic, end-to-end pipeline that transforms raw GitHub repositories into executable evaluation environments without human intervention.

Our framework introduces three key technical innovations to resolve the limitations of prior art:

1. **Neuro-Symbolic & Adaptive Synthesis:** Existing frameworks rely on unstructured extraction (for environments) and static regex (for logs), both of which are brittle at scale.

   - *Our Approach:* We introduce Synthesis Engines for both infrastructure and verification. For environments, we use **Template-Guided Synthesis** to populate security-hardened Dockerfiles (e.g., enforcing multi-stage builds). This hybrid approach leverages the reasoning of LLMs with the reliability of symbolic templates, achieving a 150% higher yield in Python repositories compared to baselines like *SetUpAgent*. For verification, we use Adaptive Parser Synthesis to generate custom Python parsers for heterogeneous logs, When deterministic parsers fail. This neuro-symbolic approach guarantees reproducibility and standardization across **10 languages and 3,971 repositories**.

2. **State-Differential Task Classification:** Current benchmarks struggle to distinguish between bug fixes and feature requests, often discarding instances where the pre-fix codebase fails to build.

   - *Our Approach:* We implement a **State-Differential Oracle** that compares three repository states: *Base*, *Before* (test patch applied), and *After* (full PR applied). We treat specific build failures in the *Before* state not as errors, but as semantic signals for **Feature Requests** (where tests rely on yet-to-be-implemented code). This allows us to verify both regression fixes and new feature implementations.

3. **Hint-Guided Trajectory Synthesis:** Standard training data generation (e.g., SWE-Gym) relies on passive filtering of easy tasks that agents can already solve.

   - *Our Approach:* We introduce an active textbfHint Injection Algorithm that converts "model-breaking" instances (where SOTA models fail) into executable training data. By injecting function signatures and dependency graphs as hints, we scaffold the agent to solve previously impossible tasks. Fine-tuning on just 145 of these frontier trajectories improves cross-lingual performance from 1.6% to 3.6%, compared to synthetic baselines, validating the critical utility of organic, high-difficulty data.

**Contributions:** This work makes the following contributions:

- **Scale & Diversity:** We produce the largest repository-level coding benchmark to date, covering orders of magnitude more repositories than SWE-bench, capturing long-tail build systems and coding patterns previously ignored.

- **Multilingual Automation:** Unlike prior automated tools limited to Python, our methodology successfully generalizes to **11 languages**, automating environment setup for diverse toolchains.

- **Expanded Task Scope (Bugs & Features):** We demonstrate the first automated methodology that systematically extracts **feature requests** via three-state State-Differential Classification. While prior approaches like SWE-bench achieve only 9% feature request representation.

- **Contamination-Resistant Evaluation:** By continuously scraping and processing the latest GitHub pull requests, we create a "living benchmark" of instances created *after* a model's knowledge cutoff, minimizing data contamination risks via temporal separation.

Table 1: Comparison of software engineering benchmarks and frameworks.

| Feature | SWE-bench / Multi-SWE | SWEE-bench (SetupAgent) | SWE-Smith | SWE-Flow | SWE-Fixer | SWE-Gym | SWE-Bench Atlas (Ours) |
|---|---|---|---|---|---|---|---|
| **Function** | Benchmark Dataset | Benchmark Generator | Syn. Data Gen. | Syn. Data Gen. | Solver Tool | RL Interface | **Live Benchmark Generator** |
| **Generation** | Manual Curation | Automated | Synthetic | Synthetic | Static Scrape (PRs only) | N/A | **Automated** |
| **Gen. Scope** | N/A | Container | Bug-fix pairs | Fix-test pairs | N/A | N/A | **Container, Log Parser, Trajectory** |
| **Env Strategy** | Pre-build Images | Extract cmds | Pre-build image | Pre-verified images | N/A | Pre-build images | **Auto-Synthesized** |
| **Scale** | 12 / 42 | 514 | 128 | 74 | ∼110k | 358 | **3,971** |
| **Languages** | Python / 9 | Python Only | Python Only | Python Only | Python Centric | N/A | **10 (Automated)** |
| **Task Scope** | Bug Fixes | Bug Fixes | Bug Fixes Only | TDD Incr. Dev. | Simple Bugs | Bug Fixes | **Bugs & Feature Requests** |
| **Log Parsing** | Static Regex | Static Regex | Static Regex | Static Regex | N/A | N/A | **Syn. Adaptive Parsers** |
| **Distribution** | Organic | Organic | Synthetic | Synthetic | Organic | Organic | **Organic** |
| **Freshness** | Static | Static | N/A | N/A | Static | Static | **Continuous (Living)** |

This table 1 situates SWE-Bench Atlas within the existing landscape, highlighting its unique contributions in automation, scale, diversity, and its focus on generating a benchmark with robust, reproducible environments while addressing the core limitations of prior work.

## 2  RELATED WORK

We review related work primarily through the lens of the structural limitations our benchmark aims to address. While **SWE-bench** and its manually curated variant **SWE-bench Verified** (Chowdhury et al., 2024) established the gold standard for evaluating LLMs on repository-level tasks, their static and manually intensive nature creates fundamental barriers to scaling software engineering evaluation.

**Scalability**  First, **synthetic generation** approaches like **SWE-Smith** and **SWE-Flow** utilize LLMs to *synthesize* training signals—either by injecting bugs into existing codebases or by inferring incremental steps from Test-Driven Development (TDD) patterns. While valuable for training efficiency, these synthetic tasks often lack the noise, ambiguity, and "in-the-wild" distribution of real human-written code. Second, **static data scaling** efforts like **SWE-Fixer** (Xie et al., 2025) aggregate massive datasets by scraping GitHub history. However, it operates as a retrieval-based pipeline without execution environments, prioritizing raw volume over execution-based verification. Third, **compute scaling** frameworks like **SWE-Gym** Pan et al. (2025) transform existing benchmarks into reinforcement learning (RL) environments to generate millions of agent trajectories. While this scales experience, it remains bound to the limited problem set of the original manually curated datasets. Finally, attempts to automate organic task collection, such as **SWEE-bench** (Vergopoulos et al., 2025), utilize agents (e.g., *SetUpAgent*) to scaffold environments but remain restricted to Python and lack support for feature requests.

**Data Contamination and "Live" Evaluation**  Static benchmarks are highly vulnerable to data contamination, as instances created before a model's knowledge cutoff are frequently memorized during pre-training. The community has responded with "live" benchmarks such as **SWE-bench-Live** and **LiveCodeBench**, which continuously harvest new problems (Zhang et al., 2025b; Jain et al., 2025). Similarly, **SWE-bench Pro** attempts to mitigate contamination by incorporating private, commercial repositories Deng et al. (2025). However, these solutions often rely on specific language ecosystems or lack the fully automated, multi-stage verification pipeline required to scale beyond hundreds of tasks to thousands.

**The Weak Test Oracle Problem**  Standard evaluation protocols assume that passing a developer-written test suite equates to a correct fix. However, this "test oracle" is often unreliable. Empirical studies on **SWE-bench** have revealed that a significant percentage of plausible patches—those that pass the provided tests—are semantically incorrect or diverge from the ground truth. This oracle limitation can lead to overestimation of model capabilities, highlighting the need for more rigorous, state-based verification methods.

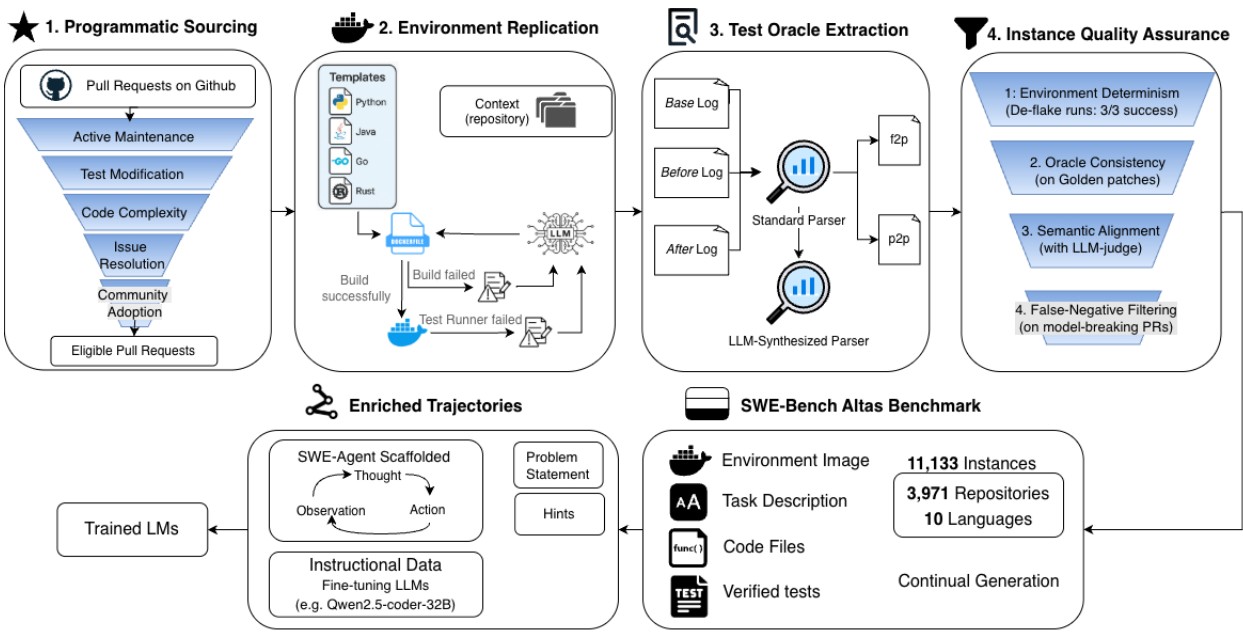

Figure 1: The framework of SWE-Bench Atlas.

**Environment Reproducibility Challenges**  Accurately reconstructing historical development environments is a primary scalability bottleneck. Complex dependency trees and version mismatches often lead to "environment rot," where test failures result from configuration errors rather than faulty model patches. While agentic systems like **SetUpAgent** (Vergopoulos et al., 2025) automate some aspects of environment creation through command extraction from documentation, they often lack the reliability of deterministic, template-guided scaffolding, restricting their success rate across polyglot repositories.

**Solution Leakage and Ambiguity**  Finally, the quality of problem statements in existing benchmarks varies significantly. A recent analysis by **SWE-bench+** (Aleithan et al., 2024) revealed that **32.67%** of successful agent resolutions involved "solution leakage," where the correct code or a direct pointer was explicitly present in the issue description or comments. This allows agents to "solve" tasks via information retrieval rather than reasoning, further skewing leaderboard rankings.

In the next section, we detail how **SWE-Bench Atlas** addresses these limitations via a pipeline that ensures scale, diversity, and rigorous verification.

## 3 METHODOLOGY: THE SWE-BENCH ATLAS FRAMEWORK

The **SWE-Bench Atlas** framework operates as a sequential pipeline designed to transform raw GitHub activity into rigorous, verifiable software engineering tasks. The architecture consists of four automated stages: programmatic sourcing, neuro-symbolic environment synthesis, state-differential oracle extraction, and automated quality assurance.

### 3.1 STAGE 1: PROGRAMMATIC SOURCING

The pipeline begins with a broad search to identify candidate tasks that represent realistic software maintenance and evolution. We employ scalable filters to process the GitHub firehose, selecting repositories and pull requests (PRs) that meet the following criteria: (a) active maintenance histories with recent commit activity; (b) evidence of community adoption (e.g., > 100 stars) and a recognizable testing framework; (c) substantial complexity, defined by codebases exceeding 10k lines of code; (d) merged PRs that explicitly resolve an issue (ensuring a link between a natural language problem description and a coding solution); and (e) PRs that include edits or additions to test files. This coarse filtering casts a wide net, identifying millions of potential candidates for the compute-intensive stages that follow.

## 3.2 STAGE 2: NEURO-SYMBOLIC ENVIRONMENT SYNTHESIS

Once a high-quality PR is identified, the next critical step is to create a fully reproducible execution environment that precisely mirrors the state of the repository at the time the issue was present. Our module uses a **Neuro-Symbolic Architecture**. We formalize the environment generation as a Neuro-Symbolic synthesis problem. The Symbolic component provides a rigid, verifiable skeleton (via templates) that enforces best practices (e.g., multi-stage builds). The Neural component acts as a constrained solver, deducing only the missing variable dependencies (e.g., package versions, entry points) that cannot be parsed via static analysis.

### 3.2.1 SYSTEM ARCHITECTURE: TEMPLATE-BASED SCAFFOLDING

To mitigate the security risks and logical errors inherent in generating Dockerfiles from scratch, our system utilizes a library of vetted, language-specific templates (e.g., Python, Java, Go, Rust). These templates enforce best practices, such as multi-stage builds, minimal base images, and non-root user execution, providing a secure structural skeleton. The templates contain semantic placeholders for dynamic content, including language versions, dependency installation commands, and entry points. The agent's objective is to populate these placeholders, thereby combining the robustness of a human-engineered structure with the flexibility of an LLM.

### 3.2.2 LLM-POWERED ITERATIVE REFINEMENT LOOP

The core instantiation process is an iterative loop that treats the Docker build and test-runner process as an empirical validation mechanism. This workflow consists of three phases:

- **Phase 1: Repository Analysis and Plan Generation.** Unlike heuristic approaches that simply scrape document files (e.g.`README`), the synthesis engine performs deep structural analysis. The LLM is granted controlled tool access via a Model Context Protocol (MCP) server that exposes repository-level operations (`clone`, `list`, `read`). Through these deterministic calls, the synthesis engine materializes the target PR at the correct commit, traverses the project tree, and inspects build scripts and manifests verbatim. This evidence is incorporated into a structured JSON plan. This tool-augmented introspection enables precise identification of required base images, package managers, and entry points, reducing hallucinations compared to static text extraction.

- **Phase 2: Build-Feedback Loop.** The system injects the JSON values into the template and attempts a `docker build`. If the build fails, the synthesis engine captures the standard error (`stderr`) output. This error trace serves as feedback for the LLM, which generates a corrected JSON plan. This self-correction loop continues until success or a retry limit (set to 5) is reached.

- **Phase 3: TestRun-Feedback Loop.** A successful build guarantees only syntactic validity. To ensure semantic correctness, the system spins up the container and attempts to execute the test runner, verifying that it works and no tests fail due to environment misconfiguration. Only environments that successfully execute the test suite are advanced to the next stage.

## 3.3 STAGE 3: AUTOMATED STATE-DIFFERENTIAL TEST ORACLE EXTRACTION

Once the environment is ready, this stage executes the code across multiple repository states and extracts the *test oracle* through automated log parsing.

### 3.3.1 THREE REPOSITORY STATES

We define three snapshots of the codebase relative to the pull request:

- **Base:** The state of the repository at the parent commit of the PR.

- **Before:** The state after applying *only* the test patch (the modifications to test files included in the PR) to the Base commit.

- **After:** The final state after applying the full PR (both test patch and implementation code).

The pipeline runs all tests under these three states—Base, Before, and After—producing three distinct execution logs.

### 3.3.2 STATE-DIFFERENTIAL CLASSIFICATION LOGIC

Existing benchmarks primarily support **Scenario A: Regression / Bug Fix**—they assume that the `Before` state is buildable and focus solely on "Fail-to-Pass" (F2P) tests. However, based on observations from large-scale real-world datasets, we find that this definition fails to capture **Scenario B: Feature Request**.

To address this limitation, we broaden the definition of F2P to encompass both scenarios, allowing the framework to handle regression fixes and feature additions in a unified manner.

- **Scenario A: Regression / Bug Fix.** If the `Before` state builds successfully, we execute the modified tests.
  - *F2P:* Tests that fail in `Before` and pass in `After`. These represent the regression.
  - *P2P:* Tests that pass in both. These represent the constraint to not break existing functionality.
- **Scenario B: Feature Request.** If the `Before` state fails to build (due to missing symbols/dependencies introduced in the PR):
  - *F2P:* We identify newly added tests in the PR that pass in the `After` state. The build failure in `Before` serves as the confirmation that the feature was absent.

### 3.3.3 SYNTHESIZED ADAPTIVE LOG PARSING

To automatically derive the test oracle—structured evidence of which tests pass or fail—the system employs a **hybrid neuro-symbolic architecture** designed to handle diverse test frameworks and noisy outputs.

We implement a hierarchical parsing strategy. The system first attempts **deterministic symbolic parsing** (using high-precision regex) for standard frameworks (full list in Appendix B) to ensure zero-cost accuracy. When symbolic parsing fails (e.g., unrecognized formats), the system falls back to **neural synthesis**, where an LLM generates a custom Python parser. Crucially, the validity of this neural output is verified symbolically via a synthetic failure injection test.

1. **Self-Correcting Synthesis Loop:** The LLM-parser generation operates within an iterative feedback loop. The system executes the synthesized parser on a sample log; if the parser crashes or extracts an implausible number of tests, the LLM receives the error trace as feedback to refine the code. This ensures resilience against noisy or non-standard output formats.
2. **Synthetic Failure Injection (Balanced Evaluation):** To empirically validate the parser's accuracy, we synthesize "Failing Test Patches." An LLM rewrites assertions within the code diff to force a test failure while preserving the patch's original structure. This creates a balanced dataset of `PASS` and `FAIL` outcomes, allowing us to verify that the synthesized parser correctly discriminates between states and does not yield false positives.

## 3.4 STAGE 4: AUTOMATED INSTANCE QUALITY ASSURANCE & VERIFICATION

To ensure the reliability of the generated instances, we deploy a four-layer **Automated Quality Assurance (AutoQA)** pipeline designed to identify and reject unstable environments, flaky tests, and ambiguous problem statements.

**Layer 1: Environment Determinism (Build Stability)** Repository-level benchmarks often suffer from transient dependency failures. We enforce **Environment Stability** by subjecting every dockerized environment to a "de-flake" process. Each instance is built and instantiated three separate times. Only instances that successfully initialize the testbed in **3/3 trials** are retained, preventing downstream evaluation noise from intermittent build failures.

**Layer 2: Oracle Consistency (Test Determinism)** To confirm that the test suite yields identical outcomes across runs, we validate **Test Determinism**. We execute the "golden solution" (the ground-truth patch) against the verify/regression tests three times in independent containers. We retain only those instances where the test results (Pass/Fail) are identical across all three runs, eliminating "flaky" tests that pass or fail based on timing conditions.

**Layer 3: Semantic Alignment & Automated Curation** We employ a rubric-based **LLM-Judge** C to evaluate the alignment between the problem statement and the test oracle. While instances with fundamental ambiguity and misalignment are rejected ("Low Quality"), we identify a recoverable class of "Medium Quality" instances where tests rely on implementation details not explicitly requested in the issue (e.g., new accessor methods). For these, we trigger an **Automated Curation** module: the system analyzes the code patch to extract the signatures of implicit dependencies and appends them to the problem statement as "Hints". This systematically repairs underspecified tasks, transforming them into high-fidelity, verifiable instances without human intervention. Details can be found in Appendix D

**Layer 4: False-Negative Filtering (Model-Breaking Verification)** To ensure that our benchmark accurately measures the upper bounds of model capability, we perform deep inspection on instances where state-of-the-art (SOTA) models fail to generate a solution. The goal is to distinguish between **True Negatives** (model limitation) and **False Negatives** (dataset defects). We utilize an automated **Trajectory & Log Inspection Module** that parses the execution trace of failed SOTA attempts. Instances where the model failure stems from infrastructure artifacts (e.g., unsupported tool, tool crashes), unsupported dependencies, or underspecified problem statements are flagged and removed. This ensures that the remaining "hard" instances represent genuine reasoning challenges.

**Human Verification (Verified Subset)** While the core pipeline is fully automated, we established a "Gold Standard" subset for high-precision evaluation. We recruited 82 pre-screened annotators to conduct comprehensive manual verification on the model-breaking instances retained from Layer 4, following the guidelines of SWE-bench Verified.

### 3.5 STAGE 5: TRAJECTORY ENRICHMENT VIA HINT-GUIDED SYNTHESIS

While Stages 1–4 generate the benchmark instances for evaluation, this final stage synthesizes high-fidelity instructional trajectories for training. Unlike frameworks such as SWE-Gym Pan et al. (2025), which generate data by passively filtering for trajectories where an agent naturally succeeds, SWE-Bench Atlas targets "Model-Breaking" instances that SOTA models fail to solve.

To convert these failures into valuable training signals, we introduce a *Hint-Guided Curation* algorithm:

1. **Failure Identification**: The system identifies instances where SOTA baselines consistently fail (0% pass rate)
2. **Contextual Scaffolding**: The pipeline analyzes the ground-truth patch to extract critical function signatures and dependency graphs. These are injected as "Hints" into the agent's system prompt to bound search space.
3. **Guided Resolution**: The agent retries the task with this scaffold. This mechanism raises the pass rate on these "hard" tasks from 0% to $\sim 70\%$, harvesting successful reasoning traces on problems that were previously unsolvable.
4. **Contamination Control & Outcome**: To prevent the model from learning to rely on hints, we apply a *Thought Regeneration* pass: an LLM rewrites the agent's reasoning trace to exclude hint-related keywords while preserving the logical solution path. This produces a dataset of "frontier" trajectories—instances at the exact boundary of capability—providing significantly higher information gain for fine-tuning than the "easy" instances captured by passive filtering.

## 4 EMPIRICAL VALIDATION

Our evaluation is four-fold: we first validate the generation pipeline, analyzing the yield, efficiency, and diversity of the resulting dataset (Section 4.1); we then benchmark state-of-the-art (SOTA) agents to establish difficulty baselines (Section 4.2); subsequently, we investigate the dataset's utility as a training resource via controlled fine-tuning experiments (4.3); and finally, we conduct a qualitative failure analysis on model-breaking instances (4.3.2), which is included in Appendix H.

### 4.1 EVALUTATING THE SWE-BENCH ATLAS PIPELINE AND DATASET

**Yield & Throughput Analysis** We initialized the sourcing module with **137,048** candidate Pull Requests (PRs) meeting our activity and complexity criteria. Of these, **28,513 (20.8%)** successfully passed the Neuro-Symbolic Environment Replication stage and State-Differential Test Oracle Extraction (Stage 2 and Stage 3), yielding a reproducible Docker container with successfully parsed logs. This success rate varies by language ecosystem, with **Python (41%)** and **Java (27%)** showing the highest resilience, while compiled languages like **C++ (9.5%)** present greater environmental challenges due to complex toolchain dependencies. Following the Automated Quality Assurance pipeline (Stage 4), **39%** of the dockerized instances were verified as deterministic and aligned, resulting in a final dataset of **11,133 instances**. The end-to-end processing time averages **67 minutes per instance**, dominated by the compilation and testing latency of the Docker build process.

**Dataset Distribution** The final dataset covers **3,971 unique repositories**, a two-order-of-magnitude increase over the 12 repositories in the original SWE-bench. This shift minimizes the risk of overfitting to specific project coding styles. A taxonomy analysis on a random subset of 488 instances confirmed the dataset's representative scope across the open-source landscape, covering diverse domains such as **DevTools (27.1%)**, **Infra/DevOps (18.5%)**, **Scientific**

Table 2: Dockerization success rates (Yield) by language

| Language | Python | Go | TS | JS | Ruby | PHP | Java | Rust | C++ | C# | C |
|---|---|---|---|---|---|---|---|---|---|---|---|
| **Yield** | 41.0% | 41.0% | 40.0% | 39.0% | 38.0% | 38.0% | 27.0% | 19.0% | 11.0% | 10.0% | 9.5% |

Table 3: Yield analysis across processing stages

| Stage | Count | Yield | Notes |
|---|---|---|---|
| Stage 1 (Sourcing) | 137,048 | 100% | Filtered by activity & complexity, etc. |
| Stage 2 & 3 (Replication & Parsing) | 28,513 | 20.8% | High variance: Python (41%) vs. C++ (9.5%). |
| Stage 4 (Quality Assurance) | 11,133 | 8.1% | High-fidelity deterministic instances. |

**Computing (12.9%)**, and **Data Engineering (10.7%)**, with the long tail covering AI/ML, Blockchain, and Embedded Systems. Furthermore, the evaluation of resolved rates, broken down by repository domain (as detailed in Figure 2a), showed significant differential performance among the models. This suggests that the inherent diversity of the benchmark serves as a high-fidelity diagnostic tool, moving beyond single-score metrics to reveal the specific strengths and weaknesses of advanced code-generation models. Distributions for code and issue types are provided in the Appendix E.

**Difficulty Distribution**  SWE-Bench Atlas dataset has a balanced distribution of task sizes by lines changed: 24.5% small (1–30), 45.6% medium (31–100), 22.3% large (101–300), and 7.6% very large (301+). Files changed also show breadth: 39.3% involve 2–4 files, 36.9% 5–8 files, 17.1% 9–15 files, and 6.7% 16+ files. The dataset includes challenging instances (12.2% with 200+ lines and 17.2% with 10+ files), ensuring a comprehensive evaluation framework covering quick fixes to large-scale refactors.

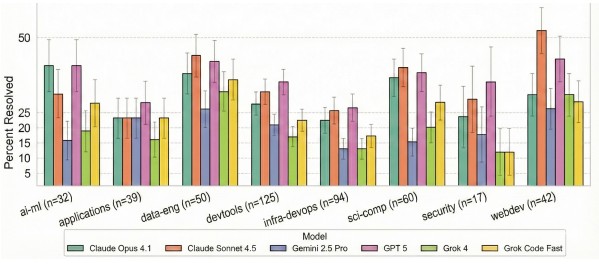
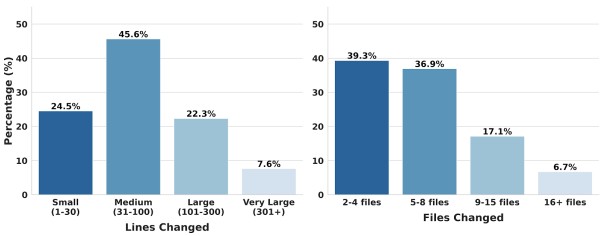

(a) Percentage of resolved repositories by type.    (b) Difficulty distribution.

Figure 2: An overall view of the dataset.

## 4.2 EVALUATING LLM AGENTS ON SWE-BENCH ATLAS

To establish performance baselines, we benchmarked leading LLM agents on the **Atlas-1,782** subset (a verified cross-lingual sample). As shown in Table 4, the benchmark presents a significant challenge even to frontier models.

- **Performance Hierarchy:** The gap between the top-performing model (**36.20%**) and mid-tier models (∼ 14%) highlights the difficulty of the benchmark.
- **Language Disparity:** Models generally perform stronger on Python and Java, likely due to the preponderance of these languages in pre-training corpora. Lower performance on Go and C++ highlights the need for multilingual benchmarks to drive progress in less represented languages.

## 4.3 FINE-TUNING EXPERIMENTS

To validate the efficacy of **SWE-Bench Atlas** as a source of high-quality training data, we conducted a controlled fine-tuning study using the **Qwen2.5-Coder 7B and 32B** variants (Hui et al., 2024). Our objective was to assess whether adding a small volume of real-world, agentic trajectories from *SWE-Bench Atlas* could improve performance on out-of-distribution polyglot tasks compared to purely synthetic baselines.

Table 4: Leaderboard of model evaluation on SWE-Agent (Jimenez et al., 2024).

| Model | Overall (Atlas-1,782k) | Per-language pass@10 | | | | | | | | |
|---|---|---|---|---|---|---|---|---|---|---|
| | | Py | Java | JS/TS | Rust | C/C++ | Go | PHP | Ruby | C# |
| gpt-5-2025-08-07 | 34.57% | **43.57%** | **41.84%** | 33.67% | 22.86% | 30.81% | 24.00% | 42.90% | 40.50% | 39.00% |
| claude-sonnet-4.5 | **36.20%** | 34.29% | 39.80% | **34.69%** | 22.86% | **57.30%** | **28.00%** | 42.90% | **53.00%** | **55.00%** |
| claude-opus-4.1 | 32.38% | 37.14% | 33.67% | 27.55% | 14.29% | 48.11% | 19.00% | 42.90% | 40.50% | 43.50% |
| claude-sonnet-4-20250514 | 31.09% | 36.43% | 31.63% | 26.53% | 17.14% | 36.22% | 20.00% | 35.70% | 37.50% | 42.50% |
| xai/grok-code-fast-1 | 30.42% | 36.43% | **41.84%** | 31.63% | 11.43% | 10.27% | **28.00%** | 50.00% | 35.00% | 42.00% |
| xai/grok-4-0709 | 25.52% | 34.29% | 38.78% | 29.59% | 22.86% | 7.03% | 17.00% | 42.90% | 34.50% | 39.00% |
| gemini/gemini-2.5-pro | 24.92% | 20.00% | 28.57% | 19.39% | 8.57% | 28.11% | 13.00% | **50.00%** | 35.50% | 34.00% |
| qwen3-coder | 24.19% | 16.43% | 31.63% | 21.43% | 14.29% | 9.19% | 19.00% | 35.70% | 39.50% | 36.50% |
| gpt-4o | 16.89% | 10.71% | 12.24% | 5.10% | 5.71% | 4.32% | 9.00% | 28.60% | 13.00% | 9.00% |

### 4.3.1 EXPERIMENTAL SETUP

We constructed distinct data mixtures to isolate the impact of trajectory source, diversity and scale. Table 4 details the composition of each dataset:

1. **Experiment 1 (Baseline):** We establish a baseline using the 5,016 synthetic trajectories from SWE-Smith. The dataset used here serves as the foundation for the SWE-Agent-LM models (7B and 32B detonated as Experiment 1a and 1b respectively).

2. **Experiment 2 (Atlas-Density):** We augment the Baseline with **179 curated trajectories** sourced from **44 GitHub issues** via *SWE-Bench Atlas*. This mixture tests the value of having multiple solution paths (higher density) for a smaller set of complex problems.

3. **Experiment 3 (Atlas-Diversity):** We augment the Baseline with **145 curated trajectories** sourced from **145 unique GitHub issues**. This mixture tests the value of maximizing issue and repository variety (higher diversity) with a comparable data volume.

4. **Experiment 4 (Atlas-Data-Scaling):** We evaluate data scaling laws by iteratively augmenting the Baseline with **200**, **400**, and **800** curated trajectories (denoted as *Atlas-Data-Scaling-1*, *Atlas-Data-Scaling-2*, and *Atlas-Data-Scaling-3*). Distinct from Experiments 2 and 3, the trajectories here utilize a hybrid human review strategy where 40% of the added data is purely synthetic (filtered only for passing the harness but with no human QA) while the remainder undergoes human review.

5. **Experiment 5 (Atlas-Model-Scaling):** We replicate the data scaling variations from Experiment 4 using the **Qwen2.5-Coder-32B** model to verify that the performance gains hold also for models of much larger sizes. Note that for the *Atlas-Data-Scaling-3 variation* (+800 SWE-Bench Atlas data) we scaled the SWE-Smith baseline to **10,032 trajectories**. This ensures a robust ratio between the specialized Atlas data and the general baseline, preventing the larger model from overfitting to the specific fine-tuning tasks at the expense of generalization (catastrophic-forgetting).

**Evaluation Protocol:** We evaluate the resulting checkpoints on **SWE-bench Multilingual** (Yang et al., 2025), a rigorous benchmark comprising 300 tasks across 42 repositories and 9 programming languages. Crucially, this benchmark contains **no Python tasks**, serving as a strict test of cross-lingual generalization. Furthermore, we ensured zero overlap between the repositories used in our *Atlas* training sets and the *SWE-bench Multilingual* test set.

**Ablation Logic: Experiment 1** functions as a reproduction of the SWE-Agent-LM performance profile, utilizing a standard fine-tuning setup in the absence of open-source configurations. **Experiments 2 and 3** disentangle the value of repository variety versus data volume, using raw, unverified trajectories to establish a strict lower-bound for data utility. **Experiments 4 and 5** confirm that gains are robust across data and model sizes; crucially, the inclusion of 40% unreviewed data verifies that our high signal-to-noise sourcing allows for scaling without being bottlenecked by manual review costs.

### 4.3.2 FINE-TUNING EXPERIMENT DETAILS

Our fine-tuning setup heavily inherits from SWE-Smith, we use the same learning rate of 5e-5, maximum 3 epochs, and max context length of 32768 and relied on the MS-Swift framework to execute fine-tuning on 8x NVIDIA H200 144G GPUs. We also follow the same XML data conversion strategy and the rejection sampling fine-tuning process. A detailed list of all the used hyper-parameters can be found in Appendix G.

### 4.3.3 RESULTS

The results of the experiment can be seen in the table below:

| Model Size | Experiment | Fine-Tuning Mixture | Performance (pass@1) | Diff CI* (95%) |
|---|---|---|---|---|
| Qwen2.5-Coder-7B | 0 | Off-the-shelf | 0 / 300 | – |
| | 1 | SWE-Smith 5k | 5 / 300 | (+1.0, +10.0) |
| | 2 | Atlas Density | 7 / 300 | (+0.0, +5.0) |
| | 3 | Atlas Diversity | **11 / 300** | (+1.0, +8.0) |
| | 4 | Atlas-Data-Scaling-1 | 6 / 300 | – |
| | | Atlas-Data-Scaling-2 | 16 / 300 | (+4.0, +16.0) |
| | | Atlas-Data-Scaling-3 | **20 / 300** | (+1.0, +8.0) |
| Qwen2.5-Coder-32B | 0 | Off-the-shelf | 4 / 300 | – |
| | 1 | SWE-Smith 5k | 12 / 300 | (+3.0, +14.0) |
| | 5 | Atlas-Data-Scaling-1 | 17 / 300 | (+1.0, +10.0) |
| | | Atlas-Data-Scaling-2 | 21 / 300 | (+1.0, +8.0) |
| | | Atlas-Data-Scaling-3* ** | **25 / 300** | (+1.0, +8.0) |

Table 5: Fine-tuning results on SWE-bench Multilingual.

* 95% confidence interval of the difference between the pass@1 performance of the current and the previous row.
** this variation uses 10,032 SWE-Smith data in the baseline mix instead of 5,016 in all the other experiments.

Incorporating just 145 Atlas trajectories (i.e., 2.8% of the mix) increased the baseline performance (from 5/300 to 11/300) and yielded a 5x increase in valid patches, demonstrating the critical value of high-diversity, "hard" multilingual samples. Notably, while the improvement from solution density (Exp 2) is marginal and borders on statistical noise (95% CI: +0.0 to +5.0), the improvement from repository variety (Exp 3) presents a robust, statistically significant signal (95% CI: +1.0 to +8.0). **These gains scale monotonically with data volume**: increasing the Atlas subset to 800 trajectories quadrupled the 7B baseline score, a trend that transferred robustly to the 32B model which reached a peak performance of 25/300.

## 5 DISCUSSION

**Limitations and future directions.** While **SWE-Bench Atlas** achieves scale through neuro-symbolic automation, we recognize that execution-based verification is a proxy for correctness, not a guarantee. Automated metrics cannot fully capture dimensions such as code maintainability or algorithmic efficiency. Furthermore, our *State-Differential Oracle* is bound by the quality of the original developer-written test suites; sparse tests may strictly allow agents to produce patches that technically pass but fail human review. While our current pipeline utilizes an LLM-Judge for semantic alignment, approximating professional maintainer standards remains a challenge. Future iterations could explore 1) scalable Human-in-the-Loop curation, leveraging community feedback or crowdsourced validation to bridge the gap between functional correctness and true patch acceptability; 2) multi-modal verification to support UI-centric or frontend-heavy tasks.

## 6 CONCLUSION

Real-world software development is messy, heterogeneous, and evolving—properties that static, manually curated benchmarks struggle to capture. While previous efforts laid the groundwork for repository-level evaluation, manual bottlenecks restricted them to a "drop in the sea" of the ecosystem. SWE-Bench Atlas addresses these limitations not merely through automation, but through a novel neuro-symbolic synthesis framework. By integrating template-guided environment scaffolding, state-differential oracle extraction, and adaptive log parsing, our approach systematically recovers complex tasks—including feature requests—across heterogeneous build systems that prior methods discard. Furthermore, our hint-guided trajectory synthesis transforms these tasks into a vital training resource for model hill-climbing. By ensuring a continuous supply of fresh, annotated problems, we mitigate leaderboard overfitting and provide infrastructure to support the development of autonomous AI systems capable of complex reasoning and self-correction.

## REPRODUCIBILITY STATEMENT

To ensure the transparency, replicability, and rigorous evaluation of our framework, we have provided comprehensive artifacts covering code, data, and experimental configurations:

- **Source Code:** We have uploaded the complete, anonymized source code for the **SWE-Bench Atlas evaluation harness** (an extension of the original SWE-bench harness). This codebase demonstrates the core Neuro-Symbolic Environment Scaffolding, State-Differential Task Classification, and Hybrid Log Parsing logic described in Section 3, enabling independent verification of our primary methodological innovations. The code includes extensive inline documentation to facilitate auditing.

- **Experimental Artifacts:** To validate the structural integrity of our generated tasks, we provide a representative subset of **500 verified benchmark instances**. These artifacts include the synthesized Dockerfiles, test patches, contextual hints, and execution logs, allowing reviewers to inspect the deterministic behavior of the pipeline.

- **Methodological Transparency:** All system prompts, heuristic thresholds, and fine-tuning hyperparameters (e.g., learning rates, epochs, hardware configuration) are documented in the **Appendix**. These are organized by pipeline stage—*Environment Scaffolding*, *Log Parsing*, *LLM-Judge*, *Hint Generation*, and *Trajectory Curation*—to allow for precise reproduction of our automated pipeline.

- **Open Science Commitment:** Upon acceptance, we are committed to establishing SWE-Bench Atlas as a live, evolving benchmark. We plan to continuously release **verified subsets** of our generated data to the public, providing the community with a fresh, contamination-resistant stream of high-fidelity tasks to drive future research in automated software engineering.

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

# A  SAMPLE INPUT AND OUTPUT IN §3.2.2

## A.1  PHASE 1: SAMPLE INPUT AND OUTPUT

**Input.**

---

**System Prompt: Expert Build Engineer**

**Role:** Expert Build Engineer AI
**Task:** Analyze a GitHub pull request and generate a JSON configuration object to populate the Dockerfile template.
**Goal:** Populate `pre_install`, `build`, `test_cmd`, and `log_parser_name` fields.

---

**Core Reasoning Steps:**

1. **Repository Inspection:**
   - **Submodules:** Detect `.gitmodules` and ensure recursive initialization.
   - **Build System Detection:** Identify primary build tools by file existence (e.g., `CMakeLists.txt` → CMake, `pom.xml` → Maven).
   - **Compiler Standard:** Parse configuration files to detect language standards (e.g., `CMAKE_CXX_STANDARD`, `sourceCompatibility`).

2. **Dependency Resolution:**
   - **System Dependencies:** Scan manifests/imports for system packages (e.g., `apt-get install`).
   - **Package Managers:** Detect/invoke language-specific managers (e.g., `vcpkg`, `pip`, `npm`, `cargo`).

3. **Build Strategy:**
   - Generate commands to clean, configure, and build the project.
   - **Constraint:** Ensure build artifacts persist in dedicated directories.
   - **Constraint:** Enforce parallel compilation (e.g., `-j$(nproc)`) for efficiency.

4. **Test Execution Logic (Crucial):**
   - **Scope filtering:** Map *modified files* in the PR to specific test targets (e.g., `pytest tests/test_mod.py`) if possible.
   - **Output Formatting:** Ensure the test runner emits machine-readable logs (XML/JSON) or verbose stdout.

5. **Log Parser Selection:**
   - Analyze test framework imports (e.g., `import pytest`, `#include <gtest.h>`) to select parser:
   - `"googletest" | "pytest" | "maven" | ... | "agentic"`

**Output Specification (JSON):**

```
{
  "pre_install": ["list", "of", "system", "commands"],
  "build": ["list", "of", "build", "commands"],
  "test_cmd": "single comprehensive test command",
  "log_parser_name": "selected_parser_id",
}
```

---

**Output.**

```
{
  "docker_specs": {
    "java_version": "17"
  },
```

```
702    "build": [
703      "mvn clean install -B -DskipTests"
704    ],
705    "test_cmd": "mvn test -B -pl project-core,project-api -am",
706    "log_parser_name": "maven"
707  }
```

## A.2 PHASE 2: SAMPLE INPUT AND OUTPUT

**Input.**

---

**System Prompt: Error Recovery & Fix**

**Context:** A previous attempt to generate a build configuration failed during execution.
**Task:** Analyze the provided error log and the original JSON configuration to identify the root cause. Based on your analysis, provide a revised JSON object that corrects the error.

**Original Failing Configuration (JSON)**

```
{
  "docker_specs": { "java_version": "17" },
  "build": [ "mvn clean install -B -DskipTests" ],
  "test_cmd": "mvn test -B -pl project-aggregator -am",
  "log_parser_name": "maven"
}
```

**Execution Error Log**

```
+ mvn test -B -pl project-aggregator -am
[INFO] Scanning for projects...
[ERROR] Could not find the selected project 'project-aggregator' in the reactor.
[ERROR] Available projects: project-parent, project-core, project-api
...
```

---

**Output.**

```
{
  "docker_specs": {
    "java_version": "17"
  },
  "build": [
    "mvn clean install -B -DskipTests"
  ],
  "test_cmd": "mvn test -B -pl project-core,project-api -am",
  "log_parser_name": "maven"
}
```

## A.3 PHASE 3: SAMPLE INPUT AND OUTPUT

**Input.**

---

**System Prompt: Test Execution Verification**

**Role:** Test Execution Validator
**Task:** Analyze the terminal output from a test command to verify if the execution was valid, complete, and semantically parseable.
**Goal:** Distinguish between a *valid test run* (where tests may fail due to code bugs) and a *broken environment* (where the runner crashes or fails to start).

---

**Core Reasoning Steps:**

1. **Execution Integrity:** Check if the test runner successfully started and completed its process without crashing or exiting early due to segmentation faults or system errors.

2. **Output Completeness:** Verify that specific test names or IDs were printed to stdout (e.g., `test_login PASSED`). This is a prerequisite for the subsequent Log Parsing stage.

3. **Scope Verification:** Confirm that the test suite ran the intended scope (all tests or the targeted subset), rather than silently exiting after a single submodule error.

4. **Failure Classification (Crucial):**
   - **Acceptable (Success):** Tests marked as `FAIL`, `SKIP`, or `XFAIL` due to logic bugs, missing credentials, or platform constraints (e.g., GPU required). These indicate a working environment. → `success:    true`
   - **Unacceptable (Failure):** Failures due to missing system libraries (`ImportError`, `ModuleNotFound`), toolchain crashes, or syntax errors in the test harness itself. These indicate a broken environment that requires rebuilding. → `success:    false`

**Output Specification (JSON):**

```json
{
  "success": true, // Set to true if environment is healthy
  "reason": "Explanation if success is false",
  "error_message": "Extracted environment error trace",
  "details": {
    "testCommandExecuted": true,
    "testNamesPrinted": true,
    "allTestsRan": true
  }
}
```

**Output.**

```json
{
  "success": true,
  "reason": "",
  "error_message": "",
  "details": {
    "testCommandExecuted": true,
    "testNamesPrinted": true,
    "mostTestsRan": true
  }
}
```

## B  STANDARD LOG PARSER UTILIZED

| Language | Log parser |
|---|---|
| Python | `pytest` — Pytest; `django` — Django test runner |
| Javascript / Typescript | `vitest` — Vitest; `jest` — Jest; `mocha` — Mocha; `karma` — Karma; `tap` — Test Anything Protocol |
| Java | `maven` — Maven Surefire; `gradle` — Gradle; `ant` — Ant |
| Go | `gotest` — standard `go test` |
| Rust | `cargo` — standard `cargo test` |
| Ruby | `rubyunit`; `minitest`; `rspec` — with JSON output transformation; `cucumber`; `tap` |
| PHP | `phpunit` |
| C/C++ | `doctest` — XML doctest; `googletest`; `catch2`; `tap` |
| C | `NUnit`; `XUnit`; `MSTest` |

## C  LLM-JUDGE FOR LAYER 3 SEMANTIC ALIGNMENT IN §3.4

### C.1  QUALITY ANALYSIS METRICS

To systematically assess PR quality, SWE-Bench Atlas computes two complementary scores—*issue_clarity* and *test_to_issue_alignment*—each ranging from 0 (best) to 3 (worst).

- **Presence of Success Criteria:** Explicit expected behavior or acceptance criteria; missing or vague criteria increase the score.
- **Specificity of Problem Description:** Concrete, unambiguous instructions reduce the score; motivational-only content, screenshots, or external links increase it.
- **Contextual Completeness:** Steps to reproduce, code snippets, or stack traces reduce the score; absence of actionable information leads to the highest score.

### C.2  QUANTIFYING TEST-TO-ISSUE ALIGNMENT

- **Core Behavior Coverage:** Tests should exercise the main functionality or bug; poor coverage yields higher scores (2–3).
- **False Negatives (Correct Solutions Rejected):** Tests too narrow; typically addressed by adding cases (e.g., score 1).
- **False Positives (Incorrect Solutions Accepted):** Missing core coverage; requires extending/modifying tests (penalized 2–3).

### C.3  PERFORMANCE BY HUMAN VERIFICATION

These results indicate the automated reviewers—intended as junior substitutes—already achieve precision near senior/lead levels ($\approx$ 0.95 vs. 0.964/0.963). Recall varies (Claude-Sonnet-4 closest to senior; GPT-5 conservative). The main gap is higher false positive rates versus humans.

## D  HINTS GENERATION PROCESS FOR LAYER 3 SEMANTIC ALIGNMENT IN §3.4

### D.1  PREDICTING WHETHER HINTS ARE NEEDED (`IS_HINT_NEEDED`)

The module predicts whether a hint is necessary (`is_hint_needed`=1 indicates needed). Decision signals:

- **Build Logs:** Detect build failures in "Before" logs via log parsing.

- **Golden Rules:** Identify elements like new function signatures to prevent avoidable failures (e.g., `new_function` vs. `newFunction`) via AST/regex.
- **LLM Judgment:** Few-shot LLM assesses whether contextual hints are required.

Evaluated on 243 senior-labeled instances; accuracy was 94.6%.

## D.2 GENERATING HINT VALUES

Generate minimal `hint_value` content focusing on essentials (e.g., critical signatures), combining golden rules with LLM judgment to maximize fairness while avoiding biasing context.

## E   `CODE_TYPE` AND `ISSUE_TYPE` DISTRIBUTIONS IN §4.1

### E.1 CODE TYPE (PRIMARY)

| code_type_primary | count | percentage |
|---|---|---|
| bug-fix | 298 | 61.0656 |
| feature | 150 | 30.7377 |
| refactor | 22 | 4.5082 |
| performance | 12 | 2.4590 |
| unknown | 3 | 0.6148 |
| dependency-update | 1 | 0.2049 |
| build-ci | 1 | 0.2049 |

### E.2 ISSUE TYPE (PRIMARY)

| issue_type_primary | count | percentage |
|---|---|---|
| bug-report | 274 | 56.1475 |
| feature-request | 188 | 38.5246 |
| performance-issue | 12 | 2.4590 |
| chore | 10 | 2.0492 |
| unknown | 2 | 0.4098 |

## F   TRAJECTORY GENERATION PROCESS IN §4.3

### F.1 DATA PREPARATION

We select **SOTA model-breaking** issues as the basis for curation, leveraging **tailored Docker environments** produced earlier in the pipeline so the agent does not need to install the environment. The scaffold supports multiple underlying LLMs with SWE-Agent. This choice is motivated by the goal of increasing **trajectory diversity**—as different models induce distinct solution paths.

### F.2 TRAJECTORY GENERATION AND SELECTION

We produce successful trajectories by engineering the system/user prompts and injecting **issue-tailored hints** that guide the agent's exploration and solutioning. Successful trajectories are identified using our extended SWE-Bench evaluation harness: a trajectory is marked as successful if its submitted solution yields successful outcomes on both **P2P** (pass-to-pass) and **F2P** (fail-to-pass) tests. Hints are crucial—combined with multiple attempts, they raise the passing rate from approximately 0% to ∼70% in our setting—enabling effective demonstration generation on SOTA-model-breaking cases.

We curate agentic trajectories that successfully resolve real-world GitHub issues, encompassing a diverse array of task types including bug fixes, feature requests, and refactoring. This approach contrasts fundamentally with synthetic

pipelines like **SWE-Smith** (Yang et al., 2025), which typically construct artificial problem constraints and focus predominantly on regression bugs.

By leveraging the *SWE-Bench Atlas* automated pipeline and the novel hint injection strategy (Appendix 3.5), we are able to harvest these high-fidelity traces to SOTA "model breaking" GitHub issues at scale without reliance on pre-existing model solutions.

We utilize **SWE-Agent** (Jimenez et al., 2024) as the agentic scaffold. Each trajectory is recorded as a structured, interleaved sequence of:

1. **Thought:** Model-generated reasoning traces (Chain-of-Thought).
2. **Action:** Executable tool invocations (e.g., `edit_file`, `run_test`).
3. **Observation:** Verbatim feedback from the environment (stdout/stderr).

This (`Thought, Action, Observation`) representation captures the dynamics of iterative problem-solving, demonstrating how an agent explores a codebase, revises incorrect assumptions, and converges on a solution. This provides significantly richer supervision than the single-shot solution prompting used in the original SWE-bench setup (Jimenez et al., 2024) or the static flows of "agentless" approaches (Xia et al., 2025). These trajectories serve as high-quality demonstrations for standard fine-tuning regimes (e.g., SFT, DPO).

### F.3 AUTOMATED TRAJECTORY QUALITY ASSURANCE

The harness alone is not a sufficient measure of quality. For instance, truncated trajectories may still pass, and hints can induce over-reliance. We therefore apply a multi-stage, automated QA pipeline:

1. **Structural validity:** remove trajectories that do not end with a final **submit** action.
2. **Hint contamination control:** remove trajectories whose **actions** or **observations** contain hint-related keywords.
3. **Thought regeneration:** prompt a language model to rewrite **thoughts** containing hint-related keywords (the prompt excludes the hints).
4. **Automated judging:** use a language-model-based evaluator to retain only high-quality trajectories.

The final automated stage (4. above) scores trajectories along four dimensions: (i) successful reproduction of the problem (for bug-fix cases), (ii) plausibility of the proposed solution, (iii) evidence of validation, and (iv) adherence to sound engineering practices.

### F.4 HUMAN REVIEW

Trajectories that pass automated thresholds undergo **expert evaluation** (HITL – human-in-the-loop). Reviewers detect non-trivial over-reliance on hints and systematic failure modes (e.g., a model consistently neglecting specific tools). They may regenerate thoughts, prune steps, or discard trajectories when warranted, focusing on quality aspects that cannot be reliably handled by algorithmic checks.

**Outcome.** The result is a scalable, HITL-augmented curation process that yields agentic demonstrations solving **SOTA-model-breaking** issues. By encoding both reasoning and interaction dynamics, these trajectories are expressly designed for **fine-tuning** code-capable language models.

## G FINE-TUNING EXPERIMENT DETAILS IN §4.3

As in the main paper, we follow the SWE-Smith setup (Yang et al., 2025). Key hyperparameters:

```
train_type full
torch_dtype bfloat16
num_train_epochs 3
per_device_train_batch_size 1
learning_rate 5e-05
weight_decay 0.01
```

```
lr_scheduler_type cosine
warmup_ratio 0.05
gradient_accumulation_steps 1 # 8 GPUs => effective batch-size = 8
max_length 32768
truncation_strategy 'right'
```

# H §4.4 CASE STUDIES IN FAILURE ANALYSIS OF SOTA MODELS

## H.1 CASE 1 — YAMLLINT: MISINTERPRETING RAW VS. PREPROCESSED TOKENS

**Task Context**

- Repository: `adrienverge__yamllint-746` (https://github.com/adrienverge/yamllint/pull/746)
- Language: Python
- Domain: Developer Tools
- Issue Type: Bug Fix (bug-report)

**Issue description from PR**

> On double-quoted multiline strings, quotes aren't needed if lines are broken on spaces, e.g.: `multiline: "this is a sentence cut into words"`. But quotes are needed when at least one line ends with a backslash (\), meaning the next spaces should be removed: `multiline: "https://example.com/a/very/very\ /very/very/long/URL"`.

**SOTA Model Erroneous Reasoning Trace (Claude Sonnet 4.5)**

> "The issue is in the `_quotes_are_needed` function. When this function checks if quotes are needed, it creates a test YAML string like `key: <string>` and tries to parse it. The problem is that when the original string was a multiline quoted string, the `token.value` contains the final parsed value (without backslashes), not the original form from the file. Let me create a test to better understand how tokens work with multiline strings: …"

The model assumed tokens were preprocessed, leading to:

```
# Quotes needed for multiline strings with line continuations (backslashes)
if is_multiline_string:
    return True
```

**Diagnosis by Our Framework**

`yamllint` reads raw token buffers; backslashes are preserved. A helper inspects raw buffers:

```
def _has_backslash_on_at_least_one_line_ending(token):
    if token.start_mark.line == token.end_mark.line:
        return False
    buffer = token.start_mark.buffer[
        token.start_mark.index + 1:token.end_mark.index - 1]
    return '\\\n' in buffer or '\\\r\n' in buffer
```

## H.2 CASE 2 — CHANGEDETECTION.IO: MISHANDLING MULTILINE REGEX AND ZERO-WIDTH MATCHES

**Task Context**

- Repository: `dgtlmoon__changedetection.io-2889` (https://github.com/dgtlmoon/changedetection.io/pull/2889)
- Language: Python
- Domain: Applications
- Issue Type: Feature Request

**Issue description from PR**

> Support multiline regex in text filtering. The current function iterates per line:
> ```
> for line in content.splitlines(keepends=True):
> ```

Rework to use `re.finditer`/`re.findall` over the whole content.

**SOTA Model Erroneous Reasoning Trace (Claude Sonnet 4.5)**

Tried span-overlap heuristics:

```
# Check if this line overlaps with any ignored range
for range_start, range_end in merged_ranges:
    if not (line_end <= range_start or line_start >= range_end):
        ignored_line_numbers.append(line_num)
        break
```

Failed a unit test for zero-width multiline regex `//ms`.

**Diagnosis by Our Framework**

1. Evaluate regex over entire content with re.MULTILINE (and re.DOTALL as needed) using re.finditer.

2. Map match.start() to line numbers by counting \n.

3. Avoid overlap heuristics; compute line indices from newline offsets.

Zero-width spans break overlap checks; whole-content matching plus newline mapping fixes it.