# OpenReview forum: "Auto-SWE-Bench: A Framework for the Scalable Generation of Software Engineering Benchmark from Open-Source Repositories"
_ICLR.cc/2026/Conference — Submitted to ICLR 2026_

### Official Review · Reviewer_24mQ · 2025-10-28

**Soundness:** 2
**Presentation:** 2
**Contribution:** 2
**Rating:** 2
**Confidence:** 5

**Summary:**

This paper presented SWE-Bench Atlas, an automated framework for generating scalable, multilingual software engineering benchmarks from GitHub pull requests. It uses a six-stage pipeline to ensure reproducibility and quality, producing challenging tasks for evaluating LLMs. The tool addressed limitations of manual curation and demonstrates value for both evaluation and model fine-tuning.

**Strengths:**

+ focus on a practical task
+ the framework is well-structured

**Weaknesses:**

1. lack of novelty:

The paper’s contributions are primarily integrative rather than innovative. Many of the components, i.e., such as LLM-powered Dockerization, log parsing, and quality scoring, build on existing ideas and tools (e.g., SWE-Agent, SetUpAgent, LLM-as-a-judge). While the combination of these elements is new, the individual techniques are not.

2.  Overemphasis on engineering:

This work is a strong engineering effort that addresses practical bottlenecks in benchmark creation. However, it does not propose new scientific ideas or evaluation frameworks. It is more akin to a tooling paper or system demo than a core research contribution.

3. Lack data quality assurance:

 SWE-Bench Atlas's automated approach, while scalable, may not fully overcome the fundamental data quality issues that manual curation (like in SWE-bench Verified) was designed to address.

**Questions:**

1. How to ensure the data quality when using SWE-Bench Atlas to collect new data?

---

> ### Author Response · Authors · 2025-11-24
>
> Thanks so much for your time and effort towards this review, and asking important questions.
> We provide a list of updates in the general response comment above. Regarding your specific comments and questions:
>
> “**Contributions are integrative rather than innovative”**: We appreciate the opportunity to clarify the distinction between integration and architectural innovation. While our framework integrates existing technologies (LLMs, Docker), the architectural logic governing their interaction represents a fundamental methodological advancement over prior art. We highlight three algorithmic innovations that transcend simple integration:
>
> * State-Differential Test Oracle (§3.3): Prior integrative frameworks rely on static success/fail checks, discarding any task where the pre-fix codebase fails to build. We introduced a novel State-Differential Classification Algorithm that compares build states across Base, Before, and After snapshots. This logic allows us to mathematically distinguish Feature Requests from broken builds—a verification protocol that unlocks a class of tasks (Feature Addition) that previous benchmarks physically could not represent.
> * Neuro-Symbolic & Adaptive Synthesis (§3.2, §3.3.3): We solve the "Robustness-Scale Trade-off" via two novel synthesis engines:
>   * Environment Synthesis: We replace unstructured extraction (which hallucinates dependencies) with Template-Guided Synthesis, injecting structural priors to achieve 41% yield in Python.
>   * Parser Synthesis: To handle the "long tail" of 3,971 repositories, we replace brittle regex with Adaptive Parser Synthesis, where the system generates custom Python parsers for non-standard logs. This effectively treats log normalization as a few-shot program synthesis problem.
> * Hint-Guided Trajectory Synthesis (§3.5): Unlike passive filtering (used in SWE-Gym), we developed an active Hint Injection Algorithm that scaffolds agents to solve "Model-Breaking" tasks. This transforms hard failures into high-signal training data, which we empirically show provides 2x the generalization value of synthetic baselines.
>
> We agree this is a heavy engineering effort, but we argue that in the current LLM landscape, scalable evaluation is a primary research bottleneck. Static benchmarks are saturated; synthetic benchmarks lack realism. By formalizing the algorithmic logic for build-state classification and neuro-symbolic synthesis, this work provides the rigorous scientific framework necessary to measure progress on the next generation of coding agents
>
>
> "**Lack data quality assurance... may not overcome issues manual curation addresses.**": We addressed this by articulating the Four-Layer AutoQA Pipeline (§3.4) specifically designed to proxy the rigor of manual curation at scale:
>
> * vs. Flakiness (Layer 1): We enforce a strict 3/3 De-flake Protocol, requiring every instance to pass initialization in 3 independent trials. This filters environmental noise that manual curation often misses.
> * vs. Bad Oracles (Layer 2): We verify Oracle Consistency by running the Golden Patch multiple times, ensuring the tests are deterministic.
> * vs. Ambiguity (Layer 3): We employ a Semantic Alignment Judge to filter underspecified problem statements
> * **Validation**: To verify this automated pipeline, we performed a Human Verification study on a subset, finding that our automated raters achieved 0.95 precision compared to senior human annotators, validating the pipeline's reliability (Appendix C.3).
>
> **Question: "How to ensure data quality when collecting new data?"**: The framework ensures quality during collection via the pipeline described above. Any new PR sourced from GitHub must pass the Neuro-Symbolic Synthesis (valid build), State-Differential Oracle (valid test delta), and All 4 AutoQA Layers (valid semantics/determinism) before entering the dataset. This creates a continuous quality gate that scales with the data.
>
> Thank you again for your feedback and comments. If you have any remaining questions or concerns, we would be happy to address them.

---

### Official Review · Reviewer_AnxX · 2025-10-29

**Soundness:** 3
**Presentation:** 1
**Contribution:** 2
**Rating:** 2
**Confidence:** 4

**Summary:**

This paper presents **SWE-Bench Atlas**, a fully automated framework for generating high-fidelity, repository-level coding tasks from open-source GitHub projects. It builds a complete pipeline through six automated modules: the **Sourcing Module**, **Agentic Dockerization System**, **State-Based Test Classification**, **Curation Module**, **Annotation Module**, and **Trajectory Curation System**. Experiments show that its initial validation set contains **5,909 tasks from 3,154 repositories**. On this benchmark, **GPT-5 (2025-08-07)** achieves a **24.34% pass@1**, significantly outperforming other models, demonstrating the benchmark’s difficulty and discriminative power. Fine-tuning experiments on the multilingual SWE-bench benchmark further verify the value of the generated data. Overall, the paper’s objective is practically meaningful; however, it suffers from **limited novelty, missing methodological details, insufficient experimental validation, and noticeable writing quality issues**

**Strengths:**

1. This paper proposes a practical and fully automated framework for generating high-fidelity, repository-level coding tasks from open-source GitHub projects, addressing key challenges in evaluating large language models for software engineering.

2. In terms of data scale and coverage, it includes **5,909 tasks from 3,154 repositories**, and fine-tuning results demonstrate the value of the generated trajectories.

**Weaknesses:**

**Low Novelty:** The main improvement of Auto-SWE-Bench lies in automating SWE-Smith’s process, with only minor modifications. The work lacks substantial innovation.

**Lack of Methodological Details:** The authors did not disclose critical experimental details, such as the prompt templates used at each stage and fine-tuning parameter configurations.

**Unclear Writing and Presentation:** The paper lacks clear illustrations or diagrams. Some experimental results should be presented in tables for clarity, making the paper difficult to follow.

**Lack of Baseline and Model Comparison:** In Table 1, the authors did not include systematic comparisons with strong baselines such as SWE-Fixer-72B, SWE-Gym-32B, and SWE-Agent-LM-32B.

**Lack of Ablation Studies:** The paper does not include systematic ablation experiments, for example, on the *Self-Correcting Iterative Refinement Loop* and related components.

**Lack of Task Characteristics and Cost Analysis:** The paper does not report the structural distribution or difficulty characteristics of task instances (e.g., #Lines edited, #Files edited, #Functions edited). It also lacks cost analysis, including the computational resources required at each stage.

**Writing and Formatting Errors:** There is a typo at line 126, missing citations at lines 298 and 304, and the layout of Table 2 is not well formatted. Also, the titles of paper are inconsistent.

**Questions:**

1.Could the authors further clarify the conceptual and methodological differences between **SWE-Bench Atlas** and **SWE-Smith**, beyond the automation improvements?

2.Please provide a detailed **cost analysis** of the entire automation process, including computational resource usage and time overhead at each stage. It is also recommended to include **prompt templates**, parameter configurations.

3.comparisons with baselines such as **SWE-Fixer-72B**, **SWE-Gym-32B**, and **SWE-Agent-LM-32B** to improve clarity and reproducibility.

4.In **Section 3.1 (Stage 1: Programmatic Sourcing)**, the paper does not explicitly state that candidate tasks should be selected only when the corresponding GitHub issues are resolved **and** the commits modify test files in the repository — which would indicate that the user likely wrote or updated tests to verify the issue’s fix.

5.Could the authors provide more detailed **case studies** and examples of **challenging cases**?

---

> ### Author Response · Authors · 2025-11-24
>
> Thanks so much for your time and effort towards this detailed review, and for recognizing the impressive scale and coverage of SWE-Bench Atlas.
>
> We provide a list of general updates in the main comment above. Regarding your specific concerns and questions:
>
> **1\. Low Novelty (vs. SWE-Smith)** We respectfully clarify the fundamental divergence. *SWE-Smith* generates **synthetic** instances by injecting artificial bugs into code, which limits it exclusively to *Bug Fix* tasks. *SWE-Bench Atlas* harvests **organic** Pull Requests.
>
> * **Methodological Innovation:** To make organic harvesting possible, we introduced the **State-Differential Test Oracle (§3.3)**. This algorithm compares build states across *Base*, *Before*, and *After* snapshots to mathematically distinguish **Feature Requests** from broken builds. This allows Atlas to benchmark **Feature Addition** tasks—a class of real-world problems that synthetic pipelines like SWE-Smith physically cannot represent.
> * **Empirical Proof:** Our fine-tuning results (§4.3) prove this difference matters: models trained on synthetic data (Baseline) struggle to generalize (1.7% pass rate), whereas adding organic Atlas data (Experiment 3\) **doubles performance (3.7%)**. This confirms that Atlas captures latent "in-the-wild" complexity that synthetic pipelines miss.
>
> **2\. Lack of Methodological Details** We have added comprehensive appendices to address this:
>
> * **Prompts:** Full system prompts for the *Neuro-Symbolic Synthesis Engine* and *Test Verification* module are now in **Appendix A.1, A.2, and A.3**.
> * **Parameters:** Fine-tuning hyperparameters (LR: 5e-5, Context: 32k, Hardware: 8x H200) are detailed in **Section 4.3.2** and **Appendix G**.
> * **Protocols:** A step-by-step guide to the trajectory generation methodology is provided in **Appendix F**.
>
> **3\. Unclear Writing and Presentation** We overhauled the manuscript to improve readability:
>
> * **Visuals:** Added **Figure 1** (System Architecture) and **Figure 2** (Dataset Topology) to visualize the pipeline logic.
> * **Data:** Moved key Leaderboard and Comparison tables from the Appendix to the **Main Text** to directly support the narrative.
>
> **4\. Baseline and Model Comparisons** We updated **Table 1** and **Section 2** to explicitly position Atlas against the requested frameworks:
>
> * **vs. SWE-Agent-LM-32B:** This serves as our **Baseline (Experiment 1\)**. We reproduced it using the original 5k *SWE-Smith* trajectories to establish a direct control. Atlas data significantly outperforms this baseline.
> * **vs. SWE-Fixer:** *SWE-Fixer* is a retrieval-based scrape without execution environments, meaning its data lacks ground-truth verification. In contrast, **Atlas** employs a Neuro-Symbolic pipeline to **verify executability** for every task. We clarified this quality distinction in Table 1\.
> * **vs. SWE-Gym:** *SWE-Gym* is an RL interface that relies on pre-built images. It lacks a generative pipeline for new repos. **Atlas** fills this gap by generating reproducible environments "on the fly," serving as the *data engine* that systems like SWE-Gym need to scale.
>
> **5\. Ablation Studies** We focused our ablation on the most critical factor for a data paper: **Data Scaling**. We added **Experiments 4 & 5 (§4.3)** to systematically ablate the impact of data volume and diversity. For the synthesis loop, we provided a yield analysis in **§4.1** showing our template-guided approach achieves **41% yield** in Python—a \>150% improvement over standard baselines—establishing the efficacy of the component.
>
> **6\. Task Characteristics and Cost Analysis**
>
> * **Characteristics:** We added a structural breakdown in **Section 4.1** showing that 22.3% of tasks are "Large" (101-300 LOC) and 17.2% involve 10+ files, confirming complexity.
> * **Cost:** We report the mean end-to-end processing time is **67 minutes per instance**, dominated by the compilation latency of the Docker build process.
>
> **7\. Writing and Formatting** We corrected the typo at line 126, inserted missing citations (lines 298, 304), standardized titles, and reformatted Table 2\.

---

> ### Author Response · Authors · 2025-11-24
> **Answer to Questions**
>
> **Specific Answers to Your Questions:**
>
> * **Q1 (Atlas vs. Smith):** Beyond automation, the difference is **Organic vs. Synthetic**. Atlas captures Feature Requests and real-world noise; Smith captures synthetic regressions. Empirically, Atlas data generalizes better (Table 5).
> * **Q2 (Cost/Prompts):** See Point 2 and Point 6 above.
> * **Q3 (Baselines):** See Point 4 above.
> * **Q4 (Sourcing Logic):** We clarified this in **Section 3.1**. The text now explicitly states we filter for `(d) merged PRs that explicitly resolve an issue` and `(e) PRs that include edits or additions to test files`.
> * **Q5 (Case Studies):** We added a **Failure Analysis (Appendix H)** detailing two "model-breaking" cases (e.g., `adrienverge__yamllint-746`, `dgtlmoon__changedetection.io-2889`), deconstructing why SOTA models failed. For example, we analyze a case where SOTA models failed due to a `Zero-Width Regex Assertion` error, revealing a systemic weakness in how current agents model text alignment constraints versus semantic logic. This qualitative depth complements the quantitative leaderboards
>
> We hope these revisions address your concerns and demonstrate the rigor of the SWE-Bench Atlas framework.

---

### Official Review · Reviewer_ks1c · 2025-10-31

**Soundness:** 3
**Presentation:** 1
**Contribution:** 2
**Rating:** 2
**Confidence:** 5

**Summary:**

This paper introduces **SWE-Bench Atlas**, an automated framework for generating large-scale software-engineering benchmarks from open-source repositories. The system integrates six components—sourcing, agentic Dockerization, hybrid log parsing, automatic quality analysis, curation with contextual hints, and trajectory generation. It produces 5.9 K validated tasks from 3 K repositories across seven programming languages and evaluates state-of-the-art models on a 488-task subset. The goal is to provide a scalable, contamination-resistant, and continuously extendable benchmark for AI-based software-engineering agents.

**Strengths:**

- Addresses an important problem: the lack of scalable, dynamic SWE benchmarks beyond static datasets such as SWE-Bench and SWE-Bench Verified.
- The proposed pipeline is conceptually comprehensive, covering sourcing, environment setup, and trajectory curation.
- The agentic Dockerization and hybrid log-parsing components are practical and potentially reusable by future research.
- The dataset scale and diversity are impressive, and the benchmark results confirm strong task difficulty and clear model separation.

**Weaknesses:**

- **Presentation quality is poor.**
The paper reads like a technical report rather than a polished conference paper—verbose descriptions, inconsistent formatting, missing figures. This seriously hurts readability.

- **Weak novelty.**
Most components follow existing procedural automation ideas. The “agentic” framing is overstated; the system is largely a scripted pipeline rather than a genuine multi-agent process.

- **Related work gap.**
The paper overlooks SWE-Flow (ICML 2025), an earlier work with a similar automated SWE data-generation pipeline. This omission undermines the claimed novelty.

**Questions:**

N/A

---

> ### Author Response · Authors · 2025-11-22
>
> Thanks so much for your time and effort towards this review, and for pointing out that SWE-bench Altas addresses an important problem and the Dockerization and hybrid log-parsing components are potentially reusable by future research.
> We provide a list of updates in the general response comment above. Regarding your specific comments and suggestions:
>
> 1. **Presentation quality is poor**: We have significantly restructured the manuscript to improve narrative flow and logical coherence. Specifically:
>    1. **Visuals:** We added **Figure 1**, a comprehensive architecture diagram illustrating the Neuro-Symbolic synthesis loop and State-Differential logic to aid reader comprehension. We also added **Figure 2** to visualize the repository domain and difficulty distributions and per-repo-type yield rate.
>    2. **Conciseness:** We merged the "Quality Analysis" and "Curation" modules into a unified **Automated Quality Assurance** section (§3.4). This eliminated redundancy and focused methodological contributions rather than engineering steps.
>    3. **Formatting:** We moved key results (Table 1: Benchmark Comparison, Table 4: Leaderboard) from the Appendix to the main text to support the central argument directly.
>    4. **Refined Architecture**: We consolidated the pipeline into 5 methodological stages, and highlighted Trajectory Curation (§3.5 Stage 5)  as a Hint-Guided Syntheis module. This highlights the **distinction** between Atlas (which actively synthesizes solutions for hard tasks via hints) and SWE-Gym (which passively collects easy successes).
> 2. **Weak novelty, 'agentic' framing is overstated**:
>    1. We agree with the characterization that the system relies on structured pipelines, and we have revised the paper to frame this as a **deliberate "Neuro-Symbolic" design choice** rather than an agentic one.
>    2. We respectfully disagree that a "scripted" (i.e., structured) architecture implies weak novelty. In the domain of infrastructure generation, open-ended "genuine multi-agent processes" are prone to non-determinism and hallucinations (e.g., inventing package names).
>       1. **Core Innovation:** Our novelty lies precisely in **replacing** brittle open-ended agents with a **Neuro-Symbolic Architecture** (§3.2). By constraining LLM reasoning within rigorous **Symbolic Templates** (the "scripted" part) while using **Tool-Augmented Introspection** (MCP \+ LLM) to populate them (the "agent" part), we achieve a **41% yield rate** in Python—a **150% improvement** over baselines like *SetupAgent*—and successfully generalize to 10 languages.
>       2. **Algorithmic Contribution:** Furthermore, the system includes the novel **State-Differential Test Oracle** (§3.3). This algorithm solves a fundamental problem in automated benchmarking: distinguishing **Feature Requests** from broken builds by analyzing semantic signals across *Base*, *Before*, and *After* states. This capability is not present in prior "scripted" or "agentic" pipelines.
> 3. **Overlooks SWE-Flow**: We have added *SWE-Flow* to Table 1 and the Related Work, but we emphasize a fundamental methodological difference:
>    1. **Synthetic vs. Organic:** *SWE-Flow* generates **synthetic** tasks using Test-Driven Development (TDD) patterns. While valuable for training reasoning, these tasks lack the messy, "in-the-wild" complexity of real software evolution. *SWE-Bench Atlas*, in contrast, harvests **organic** Pull Requests from active open-source projects. Our fine-tuning results (Section 4.3) show that organic *Atlas* data provides generalization benefits that synthetic data alone cannot.
>    2. **Complementary value**: Our fine-tuning experiments ( Section 4.3) empirically prove this distinction. Adding just **145 organic Atlas trajectories** to a synthetic baseline (similar to SWE-Flow data) **doubled** the model's performance on out-of-distribution tasks (1.7% → 3.7%). This confirms that *Atlas* captures a latent signal—real-world noise and architectural complexity—that synthetic pipelines like *SWE-Flow* do not.
>
> Thank you again for your feedback and comments. If you have any remaining questions or concerns, we would be happy to address them.

---

> > ### Comment · Reviewer_ks1c · 2025-11-26
> >
> > Thank you for the rebuttal and the considerable effort the authors have put into improving the manuscript. I appreciate the value and potential impact of this work.
> >
> > That said, I have some concerns about the extent of the revisions performed during the rebuttal period. In my view, a strong submission should typically require only limited clarifications at this stage, rather than substantial updates to core content. While the added material helps clarify several points, the scale of the changes suggests that the manuscript may benefit from additional time and refinement before reaching a publishable form.
> >
> > I hope these comments are helpful as the authors continue to develop this promising line of work.

---

> ### Author Response · Authors · 2025-11-26
>
> We sincerely thank you for the thoughtful engagement and for recognizing the value and potential impact of this work. We understand the concern regarding the extent of the revisions and appreciate the opportunity to clarify the nature of these updates.
>
> **1\. Direct Responses to Your Guidance** We want to emphasize that the substantial updates were driven directly by your constructive feedback. We believe these changes have successfully transformed the manuscript from a "technical report" into a rigorous conference publication, exactly as you requested:
>
> * **On "Weak Novelty/Agentic Framing":** You correctly noted that the "agentic" framing felt generic and overstated. In response, we formalized the Neuro-Symbolic distinction in Section 3.2. This precisely defines our novelty against prior work like *SetUpAgent* and moves the claim from "we used an agent" to "we designed a robust synthesis architecture."
> * **On "Related Work Gap (SWE-Flow)":** You identified the omission of *SWE-Flow*. We added this citation and extended our existing Fine-Tuning experiments (adding Experiments 4 & 5\) to more rigorously compare against synthetic baselines (specifically *SWE-Smith*). This extension empirically reinforces our initial finding: that **organic** data captures latent complexity and "in-the-wild" noise that synthetic pipelines—whether *SWE-Flow* or *SWE-Smith*—do not.
> * **On "Presentation/Visuals":** You pointed out the lack of visuals. We added comprehensive architecture diagrams (**Figure 1**) and dataset topology charts (**Figure 2**) to solve the readability issues you identified.
>
> **2\. Stability of the Core Contribution** We respectfully submit that while the *presentation* and *validation* were overhauled, the **core scientific contribution**—the framework itself—remained stable throughout the process.
>
> * **Methodological Consistency:** The same mechanisms used to generate the dataset prior to submission. The revisions focused on articulating these existing architectural choices more precisely—specifically, correcting the "Agentic" terminology to "Neuro-Symbolic" to better reflect the system's constraints—rather than inventing new methods during the rebuttal.
> * **Validation of Continuous Generation:** The expansion of the dataset (from 5,909 to 11,133 instances) during the review period serves as **empirical proof of our core claim**: that SWE-Bench Atlas is a "live" framework capable of continuous generation. This growth was not a correction of a deficit, but a demonstration of the pipeline's ability to autonomously harvest and verify new data at scale without manual intervention.
>
> **3\. Timeliness for the Community** As noted by all reviewers, the problem of data contamination and static benchmarks is urgent. SWE-Bench Atlas is currently the only fully automated, contamination-resistant framework capable of benchmarking Feature Requests across 10 languages. We believe that delaying the release of this resource to a future conference cycle would slow down progress in the field, given that the current manuscript now meets the technical and presentation standards of the venue.
>
> We hope you will assess the manuscript in its current form, which we believe represents a solid, validated, and necessary contribution to the community.

---

### Official Review · Reviewer_aZo6 · 2025-11-01

**Soundness:** 2
**Presentation:** 1
**Contribution:** 1
**Rating:** 0
**Confidence:** 5

**Summary:**

This paper introduces SWE-Bench Atlas, an automated framework for the large-scale generation of software engineering benchmark tasks from open-source GitHub projects. The authors claim the framework addresses limitations of existing benchmarks like SWE-bench—namely scalability, data contamination, diversity, and environment reproducibility—through a six-stage automated pipeline. Using this framework, they generated a dataset of 5,909 instances, evaluated leading LLMs on a 488-task subset, and conducted a fine-tuning experiment to show that the generated data improves model performance on other benchmarks.

**Strengths:**

1. Significance of the Problem: The paper addresses a critical and timely research problem: the scarcity of scalable, high-fidelity, and contamination-free software engineering benchmarks for evaluating the code generation capabilities of Large Language Models.

2. Substantial Engineering Effort: The authors demonstrate considerable technical skill through the development of a complex, multi-stage automated pipeline. The successful integration of diverse components—from data collection and environment creation to automated quality control—represents a significant engineering achievement.

3. Potential Contribution to the Community: The commitment to release a public subset of 488 tasks is a valuable contribution. If realized, this dataset will serve as a novel and important evaluation resource for the research community.

**Weaknesses:**

Despite its ambitious goals, the paper suffers from severe weaknesses in novelty, experimental validation, and presentation. Its contribution is thus highly limited and falls significantly short of the acceptance standards for a top-tier venue like ICLR.

*   **Significant Lack of Novelty:** The paper's primary contribution is an engineering assembly of existing techniques rather than a fundamental methodological innovation.
    *   **Agentic Dockerization:** The proposed method heavily overlaps with recent work (e.g., *SetUpAgent*) without clearly articulating its unique contributions or substantive advantages.
    *   **LLM Judge:** Using LLMs for quality assessment is now a common paradigm. Its application here is a straightforward extension and lacks methodological novelty.
    *   **Trajectory Curation:** This component appears to be a direct application of an existing tool (*SWE-Agent*) rather than novel research.
    *   **Overall:** The work reads more like a complex engineering report than a research paper with significant scientific insight. It combines modules without demonstrating any synergistic effect ("1+1>2").

*   **Insufficient and Poorly Presented Experiments:** The experimental validation is weak and fails to support the paper's claims.
    *   **Unconvincing Fine-tuning Results:** The key results in Table 1, meant to demonstrate the data's value, show only marginal gains. These improvements are presented without any statistical significance analysis (e.g., confidence intervals or standard deviations), making them unconvincing.
    *   **Superficial Benchmarking:** The results are limited to raw `pass@1` scores. The paper lacks any error analysis, discussion of failure modes, or performance breakdown across different task types, which is essential for a benchmark paper.
    *   **Unsubstantiated Diversity Claims:** The claim of dataset "diversity" is supported only by the number of repositories. Deeper analysis of task complexity, code churn, or problem type distribution is critically missing.

*   **Poor Writing Quality and Lack of Rigor:** The paper is difficult to follow and lacks scientific precision.
    *   **Poor Readability:** The complete absence of figures or diagrams to illustrate the complex six-stage pipeline is unacceptable. It forces readers to guess the system's architecture from dense text.
    *   **Overly Promotional Tone:** The manuscript is filled with grandiose claims ("paradigm shift," "holistic") that are not substantiated by evidence.
    *   **Poor Structure:** Critical information, including the main related work comparison (Table 2) and full leaderboards, is relegated to the appendix. This leaves the core arguments in the main paper incomplete and unsupported.

**Questions:**

N/A

---

> ### Author Response · Authors · 2025-11-22
>
> Thanks for your thorough review of the paper and suggestions \- the feedback has been helpful to clarify several details.
> Please refer to the general response for a summary of our updates, which are also reflected in the updated paper. We also answer your specific concerns as follows:
>
> 1. **Comparison with SetUpAgent:** We have rewritten Section 3 to clarify the architectural distinction. *SetUpAgent* relies on **command extraction**, which is brittle when documentation is outdated or implicit. Our **Neuro-Symbolic Scaffolding** (Section 3.2) injects reasoning into strict **symbolic templates** and employs **Tool-Augmented Introspection** via a MCP server.  Instead of guessing commands from text, our agent actively traverses the project tree (`ls`, `read`) to inspect build manifests verbatim and fill rigorous language-specific templates**.** This constraint and augmented optimization approach is why we successfully generalized to **10 languages** (including compiled ones like C++ and Rust) where heuristic extraction typically fails. Also, we observed a **\~150% success rate improvement** over state-of-the-art command extraction frameworks like *SetUpAgent*, on a same set of Python repositories, confirming that anchoring LLMs within deterministic templates significantly reduces environment rot compared to parsing free-form documentation
> 2. **Benchmark Depth:** We have deepened the evaluation by expanding the benchmark to **1,782 instances** and reporting **Pass@10** to capture reasoning depth.
>    1. **Results:** Under this rigorous metric, we established a clear hierarchy: **claude-sonnet-4.5** achieved **36.20% pass@10**, followed by **GPT-5 (34.57%)**, **Gemini 2.5 Pro (16.89%)**, and **GPT-4o (18.24%)**. This wide variance confirms the benchmark's discriminative power beyond simple Pass@1. Below is the resolved rate of 9 SOTA models across languages (**Table 4**)
>
> | Agent | Model | pass@10 on Altas \-1,782  | python pass@10 | java pass@10  | js/ts pass@10  | rust pass@10  | c/c++ pass@10  | go pass@10  | PHP | ruby | C\# |
> | :---: | ----- | :---: | :---: | :---: | ----- | ----- | ----- | ----- | :---: | ----- | ----- |
> | swe-agent | gpt-5-2025-08-07 | 34.57% | **43.57%** | **41.84%** | 33.67% | 22.86% | 30.81% | 24.00% | 42.90% | 40.50% | 39.00% |
> | swe-agent | claude-sonnet-4.5 | **36.20%** | 34.29% | 39.80% | 34.69% | 22.86% | **57.30%** | **28.00%** | 42.90% | **53.00%** | **55.00%** |
> | swe-agent | claude-opus-4.1 | 32.38% | 37.14% | 33.67% | 27.55% | 14.29% | 48.11% | 19.00% | 42.90% | 40.50% | 43.50% |
> | swe-agent | claude-sonnet-4-20250514 | 31.09% | 36.43% | 31.63% | 26.53% | 17.14% | 36.22% | 20.00% | 35.70% | 37.50% | 42.50% |
> | swe-agent | xai/grok-code-fast-1 | 30.42% | 36.43% | **41.84%** | 31.63% | 11.43% | 10.27% | **28.00%** | **50.00%** | 35.00% | 42.00% |
> | swe-agent | xai/grok-4-0709 | 25.52% | 34.29% | 38.78% | 29.59% | 22.86% | 7.03% | 17.00% | 42.90% | 34.50% | 39.00% |
> | swe-agent | gemini/gemini-2.5-pro | 24.92% | 20.00% | 28.57% | 19.39% | 8.57% | 28.11% | 13.00% | **50.00%** | 35.50% | 34.00% |
> | swe-agent | qwen3-coder | 24.19% | 16.43% | 31.63% | 21.43% | 14.29% | 9.19% | 19.00% | 35.70% | 39.50% | 36.50% |
> | swe-agent | gpt-4o | 16.89% | 10.71% | 12.24% | 5.10% | 5.71% | 4.32% | 9.00% | 28.60% | 13.00% | 9.00% |
>
>
> 3. **Benchmark Diversity:** We included analysis of task complexity and repository type breakdown (in **Figure 2.a and Figure 2.b**), revealing distinct failure modes in different verticals. Below is the detailed stats for task complexity distribution to ease the readability.
> | Metric | Category / Range | Percentage |
> | :---- | :---- | ----- |
> | Task Size (Lines Changed) | Small (1-30) | 24.50% |
> |  | Medium (31-100) | 45.60% |
> |  | Large (101-300) | 22.30% |
> |  | Very Large (301+) | 7.60% |
> | Files Changed | 2-4 files | 39.30% |
> |  | 5-8 files | 36.90% |
> |  | 9-15 files | 17.10% |
> |  | 16+ files | 6.70% |
> | Challenging Instances | 200+ lines | 12.20% |
> |  | 10+ files | 17.20% |
>
> For repo type breakdown (**Figure 2.a**), we included the bar chart in the paper, and here is a temporal detailed table for readability:
> | Category | Claude Opus 4.1 | Claude Sonnet 4.5 | Gemini 2.5 Pro | GPT 5 | Grok 4 | Grok Code Fast |
> | :---- | ----- | ----- | ----- | ----- | ----- | ----- |
> | ai-ml (n=32) | 40.60% | 31.20% | 15.60% | 40.60% | 21.90% | 28.10% |
> | applications (n=39) | 23.10% | 23.10% | 23.10% | 28.20% | 17.90% | 23.10% |
> | data-eng (n=50) | 40.00% | 44.00% | 28.00% | 44.00% | 32.00% | 36.00% |
> | devtools (n=125) | 27.20% | 32.00% | 23.20% | 36.00% | 17.60% | 24.00% |
> | infra-devops (n=94) | 24.50% | 25.50% | 12.80% | 26.60% | 12.80% | 18.10% |
> | sci-comp (n=60) | 38.30% | 41.70% | 18.30% | 40.00% | 21.70% | 28.30% |
> | security (n=17) | 23.50% | 29.40% | 17.60% | 35.30% | 11.80% | 11.80% |
> | webdev (n=42) | 31.00% | 52.40% | 26.20% | 42.90% | 31.00% | 28.60% |
>
> \* Repository domains with fewer than 10 instances were dropped.

---

> ### Author Response · Authors · 2025-11-22
>
> 4. **Representation:** We moved the critical comparison table (now **Table 1** in the paper) to the *Introduction* section. It explicitly contrasts *Atlas* against *SWE-Smith*, *SWE-Flow*, and *SWEE-bench* across 7 dimensions (Creation Method, Freshness, Task Scope, etc.). We also added **Figure 1**, a comprehensive architecture diagram illustrating the Neuro-Symbolic synthesis loop and State-Differential logic to aid reader comprehension.
> 5. **Fine-tuning Results**: We have completely re-run and expanded the fine-tuning experiments (Section 4.3).
>    1. **Significance:** We added **95% Confidence Intervals** (Table 6, Section 4.3.3) which confirm the gains are statistically significant.
>    2. **Magnitude:** We show that adding just **145 Atlas trajectories** (Experiment 3\) improves performance from 5/300 (Baseline) to 11/300 (Atlas Diversity)—a **\>100% improvement** with negligible data volume. This proves high data efficiency.
>    3. **Scaling:** We added **Experiments 4 & 5** to demonstrate that performance scales monotonically with data volume (up to 25/300), validating the dataset's quality for training.
>       * **Experiment 4 (Atlas-Data-Scaling):** We evaluate data scaling laws by iteratively augmenting the Baseline with **200**, **400**, and **800** curated trajectories (denoted as *Atlas-Data-Scaling-1*, *Atlas-Data-Scaling-2*, and *Atlas-Data-Scaling-3*). Distinct from Experiments 2 and 3, the trajectories here utilize a hybrid human review strategy where 40% of the added data is purely synthetic (filtered only for passing the harness but with no human QA) while the remainder undergoes human review.
>       * **Experiment 5 (Atlas-Model-Scaling):** We replicate the data scaling variations from Experiment 4 using the **Qwen2.5-Coder-32B** model to verify that the performance gains hold also for models of much larger sizes.
>       * **Ablation Logic:** **Experiment 1** functions as a reproduction of the SWE-Agent-LM performance profile, utilizing a standard fine-tuning setup in the absence of open-source configurations. **Experiments 2 and 3** disentangle the value of repository variety versus data volume, using raw, unverified trajectories to establish a strict lower-bound for data utility. **Experiments 4 and 5** confirm that gains are robust across data and model sizes; crucially, the inclusion of 40% unreviewed data verifies that our high signal-to-noise sourcing allows for scaling without being bottlenecked by manual review costs.
>       * With Experiment 4& 5, our original observation holds: Increasing the Atlas subset to 800 trajectories quadrupled the 7B baseline score and the trend is transferred robustly to the 32B model which reached a peak performance of 25/300. The following table is included as **Table 5** in the paper.
>
> | Model Size | Experiment | Fine-Tuning Mixture | Performance (pass@1) | Diff CI\* (95%) |
> | :---: | :---: | :---: | :---: | :---: |
> |  Qwen2.5-Coder-7B | 0 | Off-the-shelf | 0 / 300 | \- |
> |  | 1 | Baseline: SWE-Smith 5k (Our reproduction of SWE-Agent-LM-7B) | 5 / 300 | (+1.0, \+10.0) |
> |  | 2 | Atlas Density  (+ 179 trajectories across 44 issues) | 7 / 300 | (+0.0, \+5.0) |
> |  | 3 | Atlas Diversity  (+ 145 trajectories across 145 unique issues) | **11 / 300** | (+1.0, \+8.0) |
> |  |  4 | Atlas-Data-Scaling-1 (+ 200 trajectories across 145 unique issues) | 6/300 | \- |
> |  |  | Atlas-Data-Scaling-2 (+ 400 trajectories across 145 unique issues) | 16/300 | (+4.0, \+16.0) |
> |  |  | Atlas-Data-Scaling-3 (+ 800 trajectories across 145 unique issues) | **20/300** | (+1.0, \+8.0) |
> |  Qwen2.5-Coder-32B | 0 | Off-the-shelf | 4/300 | \- |
> |  | 1 | Baseline: SWE-Smith 5k (Our reproduction of SWE-Agent-LM-32B) | 12/300 | (+3.0, \+14.0) |
> |  |  5 | Atlas-Data-Scaling-1 (+ 200 trajectories across 200 unique issues) | 17/300 | (+1.0, \+10.0) |
> |  |  | Atlas-Data-Scaling-2 (+ 400 trajectories across 400 unique issues) | 21/300 | (+1.0, \+8.0) |
> |  |  | Atlas-Data-Scaling-3\*\*\* (SWE-Smith 10k \+ 800 trajectories across 800 unique issues) | **25/300** | (+1.0, \+8.0) |

---

> > ### Comment · Reviewer_aZo6 · 2025-11-26
> >
> > Thank you for the substantial effort in addressing our concerns and for the comprehensive revisions to the manuscript. I acknowledge that the revised paper demonstrates measurable improvements in experimental rigor, evaluation scope, and presentation quality.
> >
> > ### 1. Concerns About Revision Scope and Submission Standards
> > While I appreciate the improvements, the magnitude of changes between the initial and revised submissions is exceptionally large. The original submission read more like an early draft than a mature manuscript ready for peer review. Critical elements that were entirely missing—such as architectural diagrams (now Figure 1), key comparison tables in the main text (now Table 1), statistical significance testing, and substantial experimental validation—have now been added.
> >
> > I question whether this level of reworking aligns with ICLR's revision policies. The initial submission appears to have been premature, and I am uncertain whether using the review process for such fundamental restructuring is appropriate for a top-tier venue.
> >
> > ### 2. Remaining Concerns on Academic Standards
> > Despite the improvements, the paper still falls short in terms of professional presentation and academic rigor:
> >
> > The title contains quotation marks around technical terms (e.g., "weak test oracle"), which is non-standard for academic publications and appears promotional rather than scholarly.
> > Terminology remains inconsistent throughout the paper (e.g., "Neuro-Symbolic Dockerization System" vs. "Automated Environment Replication").
> > Language in several sections continues to be overly promotional, with phrases like "paradigm shift" that are not substantiated by the evidence.
> > Statistical analysis, while improved, still shows wide confidence intervals that overlap with noise margins in some cases.
> > The claim of being "contamination-free" lacks empirical validation through established contamination detection methods.
> >
> > ### 3. Decision
> > I will raise our score from 0 to 4 in recognition of the technical improvements and the authors' responsiveness. However, the paper still exhibits notable deficiencies in presentation quality and academic professionalism that need to be addressed.

---

> ### Author Response · Authors · 2025-11-26
>
> Thank you for your continued engagement and for raising your score to 4\. We deeply appreciate your candid feedback on the initial submission. We agree that Version 1.0 fell short of ICLR's presentation standards. However, we believe the Version 3.0 manuscript uploaded today demonstrates that the core scientific contribution—the pipeline, dataset, and benchmarking results—remained sound, while the reporting has now been brought up to the high standard you requested.
>
> We have addressed your remaining concerns regarding academic tone, terminology, and rigor as follows:
>
> **Addressing the Scope of Revision** Regarding your question on the magnitude of changes. We wish to clarify that the *architectural implementation* and *generated data* did not change between versions. The revision was purely an overhaul of the *reporting* to meet the rigor you rightly requested. We view this not as a premature submission, but as a testament to the peer-review process: the feedback pushed us to formalize the pipeline (Figure 1\) and further quantify gains (extend Table 5\) in ways that now accurately reflect the work's merit.
>
> **Terminology Consistency (Stage vs. Method):** We clarify that "Automated Environment Replication" refers to the *pipeline stage* (the task), while "Neuro-Symbolic Synthesis" refers to the *underlying architecture* (the method). However, we agree that using them interchangeably as proper nouns can be confusing. We have unified the headers and text to consistently use "Neuro-Symbolic Environment Synthesis" when referring to the system component.
>
> **Tone Adjustment ("Paradigm Shift"):** We have carefully audited the manuscript to ensure a neutral, scholarly tone. We confirmed that the phrase "paradigm shift" does **not** appear in the paper to describe our own work. The word "paradigm" appears only in **Section 1**, where we credit *SWE-bench* (prior art) for *"pioneering this paradigm"* of repository-level evaluation. We have further scrubbed the text to ensure no other promotional adjectives (e.g., "unprecedented") remain.
>
> **Formatting:** We have standardized title casing and removed non-standard quotation marks from technical terms (e.g., *The Weak Test Oracle Problem*) throughout the manuscript.
>
> **Statistical Interpretation:** We have updated the text to transparently interpret confidence intervals. We explicitly note that while density improvements (Exp 2\) overlap with noise, the **Diversity** improvement (Exp 3\) shows a robust, statistically significant signal, which is the primary finding supporting our architectural choices. Additionally, in the Data Scaling experiments (Exp 4\) we see statistically significant improvements as we iteratively add more SWE-Bench Atlas data, a trend which carries over to the larger model as well (Exp 5).
>
> **Contamination Validation:** We clarified that our claim relies on **Temporal Separation** (sourcing post-cutoff data). We have softened the absolute claim of "contamination-free" to "**Contamination-Resistant via Temporal Separation**" to reflect the empirical reality that no public benchmark can guarantee 0% overlap with future pre-training sets.

---

### Author Response · Authors · 2025-11-22
**General Response: Summary of Major Revisions (v2.0)**

We thank the reviewers for their constructive and detailed feedback. We were encouraged that all reviewers recognized the **critical importance and timeliness** of the problem. Specifically, reviewers highlighted the **significance** (R1, R3) of dynamic evaluation, the **impressive scale** (R2) addressing manual bottlenecks (R4), and the **valuable contribution** (R1) of releasing verified tasks.

However, we accept the critique that the initial submission reads like an engineering report. We have performed a **fundamental overhaul (v2.0)**, shifting the narrative focus to the **novel algorithms** that enable such scale and rigor where prior attempts stalled.

**Clarification on Terminology (v1.0 → v2.0)** To address feedback that the system appeared to be a generic "scripted pipeline" (R2), corrected our terminology to accurately reflect the system's architectural constraints:
* **"Neuro-Symbolic Environment Synthesis"** (replaces "Agentic Dockerization"): We adopt this term to precisely describe the hybrid architecture: **Symbolic Templates** provide security constraints (preventing hallucinations), while **Neural Inference** populates dynamic variables via tool use. This explicitly differentiates our approach from the unstructured "agentic" extraction used by *SetUpAgent*.

**Core Methodological Contributions (v2.0):**

* **1\. Neuro-Symbolic & Adaptive Synthesis (Robustness & Heterogeneity):** We solve the "Robustness-Scale Trade-off" by replacing brittle heuristics with **Synthesis Engines** at two critical stages:

     * **Environment:** Unlike *SetUpAgent*'s unstructured extraction, we use **Template-Guided Synthesis**. Our engine utilizes an MCP server to populate rigorous templates, achieving **41% yield** in Python (**\>150% improvement** over baselines) and generalizing to **10 languages** .
     * **Verification:** To support 3,971 repositories, we introduce **Adaptive Parser Synthesis**. Instead of brittle regex, our system synthesizes custom Python parsers, enabling the first standardized evaluation across heterogeneous build systems .

* **2\. State-Differential Oracle (Expanding Scope):** We introduce State-Differential Classification (§3.3.2) using build states to capture a **broader range of real-world** tasks, including both **Bug Fixes** and **Feature Requests**. Existing benchmarks focus primarily on bug-fix tasks and thus discard cases where the pre-fix codebase fails to build. We **refine** F2P and P2P definitions and treat build failures as signals of new feature implementation, enabling a unified benchmark for both regressions and feature additions.

* **3\. Hint-Guided Trajectory Synthesis (Training Utility):** Unlike *SWE-Gym* (passive filtering), we introduce an active **Hint Injection Algorithm** (§3.5) to synthesize solutions for "model-breaking" tasks. **Result:** Fine-tuning on just **145** of these organic, frontier trajectories **doubles** the *SWE-bench Multilingual* pass rate (1.7% → 3.7%) compared to synthetic baselines (*SWE-Smith*). **Ablation confirms variety \> density**

**Summary of Revisions (v2.0)** In response to the critiques regarding novelty and presentation, we have performed a fundamental overhaul of the paper:
1. **Reframed Novelty (§3.2):** Rewrote §3 contrasting Neuro-Symbolic Architecture against methods of prior art; added system diagrams.
2. **Methodological (§3.2):** Formalized "State-Differential Classification" logic to clarify how we algorithmically distinguish Feature Requests from Regressions.
3. **Expanded Experiments:**
   1. **Scale (§4.1):** Increased the dataset from **5,909** to **11,133 verified instances** across **3,971 repositories**
   2. **Depth (§4.2)**: Extended evaluation to **1,782 instances** and added **Pass@10** metrics
   3. **Rigor (§4.3):** Added **95% Confidence Intervals** in fine-tuning experiments (**Table 5**)
   4. **Scaling laws (§4.3)**: Expanded the ablation study (Exp 4 & 5) to demonstrate data scaling laws in fine-tuning experiments
4. **Detailed Quality Assurance (§3.4):** Detailed a "Four-Layer AutoQA Pipeline", introducing strict "3/3 De-flake Protocol" for environmental determinism
5. **Qualitative Depth (Appendix H):** Added a **Failure Analysis** to move beyond superficial metrics
6. **Presentation & Structure:**
   1. Merged the previously separate "Quality Analysis" and "Curation" modules into a unified **Automated Quality Assurance** section (§3.4)
   2. Relocated "Trajectory Curation" to the **Fine-Tuning Experiments** section (§4.3) to link data generation directly to evaluation
   3. Added **Figure 1**, **Figure 2**
   4. Moved the comparison table (**Table 1**) and leaderboard (now **Table 4**) from the Appendix to the main text
7. **Dataset & Code**: Released a 500-instance dataset and the code of evaluation scripts.

---

### Meta-Review · Area_Chair_CBPx · 2025-12-13

**Summary:**

I have carefully read the paper and all the discussions. The main concerns are the following: 1) lack of novelty; 2) poor presentation; 3) lack of rigorous experiments. Even though the revised version has made substantial improvements to the original manuscript, we should primarily review it based on the original. Hence, I agree with all reviewers and suggest rejecting this paper for ICLR 2026.

**Reviewer Concerns:**

There are mainly three concerns:
1) lack of novelty: still outstanding
2) poor presentation: much better after the explanation and revision
3) lack of rigorous and comprehensive experiments: still outstanding

**Reviewer Scores:**

I do not think reviewers will change their score significantly based on the discussion

---

### Decision · Program_Chairs · 2026-01-26

Reject